# Optimal Regularization for Performative Learning

## Abstract

In performative learning, the data distribution reacts to the deployed model—for example, because strategic users adapt their features to game it—which creates a more complex dynamic than in classical supervised learning. One should thus not only optimize the model for the current data but also take into account that the model might steer the distribution in a new direction, without knowing the exact nature of the potential shift. We explore how regularization can help cope with performative effects by studying its impact in high-dimensional ridge regression. We show that, while performative effects worsen the test risk in the population setting, when moving to the over-parameterized regime where the number of features exceeds the number of samples, the optimal regularization in the presence of performativity helps reduce the variance in the estimated parameters, thereby improving performance. We show that the optimal regularization scales with the overall strength of the performative effect, making it possible to set the regularization in anticipation of this effect. We illustrate this finding through empirical evaluations of the optimal regularization parameter on both synthetic and real-world datasets.

## 1 Introduction

When machine learning predictions affect user outcomes, deployed models can induce shifts in the data distribution. These shifts may result from strategic user behavior—where individuals try to secure favorable outcomes such as loan approval or college admission (Bechavod et al., 2021; Narang et al., 2023; Wang et al., 2023)—or from self-fulfilling prophecies, for example in economic forecasts, recommendation systems, or predictive policing (Morgenstern, 1928; Ensign et al., 2018; Ursu, 2015). Such distribution shifts can undermine predictive performance over time, amplifying bias and reducing model quality (Taori & Hashimoto, 2023; Pan et al., 2024). Performative learning (Perdomo et al., 2020) addresses this feedback loop by parameterizing the data distribution with the same parameter as the model. This allows optimization to account not only for the training loss but also for the steering of the data distribution.

Unfortunately, while optimizing model parameters is a classical problem in machine learning, estimating the performative effect on the distribution is generally infeasible, as the distribution is unknown to the learner. Several algorithms have been proposed to approximate this effect (Miller et al., 2021; Izzo et al., 2022; Cyffers et al., 2024), typically by assuming that it depends on a small number of parameters in a sufficiently simple way that can be learned across the first few deployments. However, these methods are limited to relatively toy examples in small dimensions and may be impractical in real-world settings. In particular, many approaches require numerous repeated deployments, alternating between loss minimization and distribution steering. Yet in practice, deployment often happens only once after full training. This makes repeated risk minimization (RRM) (Perdomo et al., 2020)—where one trains until convergence before deployment, and the number of deployments is small—the default in many applications, even though it remains largely unaddressed by existing mitigation methods. This motivates a shift away from exact estimation of performative effects toward the study of principled choices of loss functions and models, and in particular regularization is a natural and tractable candidate.

In this work, we study how regularization mitigates performative effects in repeated retraining. Unlike estimation-based methods, regularization does not depend on a precise characterization of the

distribution shift, avoids their limitations, and introduces little computational overhead. Prior work suggests its potential benefits: Perdomo et al. (2020) proved that retraining converges to an optimal solution under assumptions tied to the strong convexity of the loss, which ridge regularization can enforce; more recently, Cyffers et al. (2024) showed that, in classification, the performative optimum can be interpreted as a regularized version of the non-performative problem, with numerical evidence that ridge penalties perform well in small-dimensional classification tasks. A natural limitation, however, is that regularization may encourage reliance on spurious features (Bombari & Mondelli, 2025), especially when such features are reinforced by performativity.

To better understand these tradeoffs, we study the role of ridge regularization in linear regression in the presence of performativity and spurious features. We consider both *(i)* the population regime, with enough data to recover exactly the unknown vector of regression coefficients at each deployment, and *(ii)* the over-parameterized regime, where the number of data samples is a fixed fraction of the number of parameters. This last setting, though simple, captures behaviors relevant to deep learning, such as double descent (Belkin et al., 2019; Hastie et al., 2022), benign overfitting (Bartlett et al., 2020) and adversarial robustness (Fawzi et al., 2018; Ribeiro et al., 2023). In performative learning, it also brings the additional advantage that parameters and data live in the same space, simplifying the encoding of performative effects. The theoretical framework we develop enables us to provide strong evidence for the effectiveness of regularization under performativity, and to show how regularization should be scaled. More precisely, our contributions are summarized below.

1. In the population regime, we characterize how the risk depends on magnitude and direction of the performative effect, as well as on spurious features (Theorem 1). We find that the optimal regularization is proportional to the strength of the performative effect and it mitigates the performance loss due to performativity: zero excess risk can be achieved with identity covariance and constant entries of the performative vector, while the risk remains significant in the presence of a complex covariance structure and highly variable entries of the performative vector (Corollary 2).

2. In the proportional regime with random data, we establish a deterministic equivalent of the performative fixed point, depending only on population covariance and regularization (Theorem 3). The analysis of this deterministic equivalent then unveils a remarkable phenomenology: for small noise variance, the optimal regularization moves in the same direction as the performative effect on the predictive features, while it moves in the opposite direction as the performative effect on the spurious features; remarkably, the optimally-regularized risk improves in the presence of a performative effect that reinforces existing trends.

3. We illustrate these behaviors both on synthetic data and real-world dataset (Housing, LSAC).

## 2 RELATED WORK

**Performative learning.** Performative learning was introduced by Perdomo et al. (2020), where retraining was also shown to converge under assumptions including strong convexity. Subsequent works demonstrate that retraining can sometimes enable adaptation over time (Li et al., 2022; Drusvyatskiy & Xiao, 2023) but also fail dramatically (Miller et al., 2021; Izzo et al., 2022; Cyffers et al., 2024). The role of model choice has not yet been studied in performative learning, although it has been examined in the related setting of collective action (Ben-Dov et al., 2024). Interestingly, Bechavod et al. (2021) observed that performativity can, in some cases, even improve performance, and our work provides further evidence supporting this claim. Performative learning effects tend to be harder to mitigate in high-dimensional settings, as noted in Jagadeesan et al. (2022); Bracale et al. (2025), motivating the study of performative learning in high dimensions.

**High-dimensional regression and role of ridge regularization.** The high-dimensional setting where the numbers of features and samples are large and scale proportionally was considered by a rich line of work: the test error of ridgeless and ridge regression is characterized in (Hastie et al., 2022; Wu & Xu, 2020; Richards et al., 2021; Tsigler & Bartlett, 2023), with Cheng & Montanari (2024) going even beyond the proportional regime; max-margin classification is studied in (Montanari et al., 2025; Deng et al., 2022), model compression in (Chang et al., 2021), distribution shift in (Patil et al., 2024; Mallinar et al., 2024), transfer learning in (Yang et al., 2025; Song et al., 2024) and learning from surrogate data in (Ildiz et al., 2025; Kolossov et al., 2024; Jain et al., 2024). The role of ridge regularization was also investigated, with Hastie et al. (2022) optimally tuning the ridge and Richards et al. (2021) giving conditions for the optimality of ridgeless interpolation. The sign of the

optimal ridge penalty was studied for the standard in-distribution regression setup (Wu & Xu, 2020; Tsigler & Bartlett, 2023), as well as out-of-distribution (Patil et al., 2024): these works give conditions under which the optimal ridge is negative, associating the phenomenon of negative optimal regularization to over-parameterization. Our paper shows that such a phenomenon occurs also in the population setting, due to performative effects. The distribution of the empirical risk minimizer was established in (Han & Xu, 2023). Leveraging this characterization, spurious correlations were studied by Bombari & Mondelli (2025) and weak-to-strong generalization by Ildiz et al. (2025). We will also build on these tools to analyze the risk of repeated risk minimization.

## 3 PRELIMINARIES AND PROBLEM SETUP

In this section, we introduce our performative regression setting. We consider a sequence of model deployments $(\theta_k)_k$, and let $\mathcal{D}(\theta)$ be the dataset generated in reaction to the deployment of $\theta$. At each deployment, $n$ new samples are collected, and the model is fully retrained. This setting, known as repeated risk minimization (RRM) (Perdomo et al., 2020), reflects real-world scenarios where deployments are costly and thus limited in number. It also aligns with the fact that convergence to the fixed point is fast and requires only a few iterations to reach equilibrium in practice. We encode the performative effect as a shift in the label, where each feature's contribution varies depending on an additional linear term in the model parameter. This is formalized below.

**Assumption 1** (Regression performative model). For $\theta \in \mathbb{R}^p$, samples from $\mathcal{D}(\theta)$ are taken i.i.d. with features $x \sim \mathcal{N}(0, \Sigma)$ drawn independently of $\theta$ and with the label $y$ given by

$$y = x^\top \theta^*_{\text{pop}} + x^\top D\theta + w, \quad w \sim \mathcal{N}(0, \sigma^2). \tag{1}$$

We assume $p = 2d$, $(\theta^*_{\text{pop}})^\top = (a^\top, 0)$ with $a$ having zero mean and covariance $I_d/d$, and $D = \text{diag}(b, c)$ where $b, c \in \mathbb{R}^d$ with $\|b\|_\infty, \|c\|_\infty < 1$.

This model generalizes the one-dimensional setting (Example 2.2) in Perdomo et al. (2020), where labels follow a binomial distribution with parameter $\frac{1}{2} + x\theta^* + x\bar{b}\theta$, for $\theta^* \in (0, \frac{1}{2})$ and $\bar{b} < \frac{1}{2} - \mu$. Focusing on label shifts is natural in regression: it keeps the feature distribution centered and unchanged across deployments despite performative effects, and it can always be enforced through pre-processing.

The performative term $x^\top D\theta$ enforces coordinate-wise effects. This is consistent with previous work (Cyffers et al., 2024; Izzo et al., 2022; Hardt & Mendler-Dünner, 2023) and also close to the model $y = x^\top \theta^*_{\text{pop}} + \mu^\top \theta + w$ studied by Miller et al. (2021), where the performative effect does not depend on $x$ but only on a fixed vector $\mu$. Assuming linearity in $\theta$ is reasonable, as performative effects are expected to be moderate to avoid iterations to diverge. We specify in the rest of the paper when the fact that $D$ is diagonal is needed. Intuitively, diagonal coefficients can be interpreted directly as the modifications made by a strategic agent, depending on how the feature is used and the cost of modifying it. Most existing methods impose explicit constraints on the performative effect (Miller et al., 2021), and our setting is no more restrictive: we only require $\|b\|_\infty, \|c\|_\infty < 1$. We set the second half of $\theta^*_{\text{pop}}$ to zero to represent spurious features, and $c$ captures the corresponding performative effect. This enables us to express correlations between predictive and spurious features via the block structure of the covariance

$$\Sigma = \begin{bmatrix} \Sigma_1 & \Sigma_{12} \\ \Sigma_{12} & \Sigma_2 \end{bmatrix}, \tag{2}$$

where $\Sigma_1$ denotes the covariance for the predictive part, $\Sigma_2$ the covariance for the spurious part, and $\Sigma_{12}$ the covariance between the two blocks. Under this setting, RRM corresponds to solving

$$\theta_k = \arg\min_{\theta \in \mathbb{R}^p} \left\{ \frac{1}{2n} \sum_{i=1}^n \ell\left(x_i^{(k-1)}, y_i^{(k-1)}; \theta\right) + \frac{\lambda}{2}\|\theta\|_2^2 \right\}, \tag{3}$$

where $\{(x_i^{(k-1)}, y_i^{(k-1)})\}_{i=1}^n \overset{\text{i.i.d.}}{\sim} \mathcal{D}(\theta_{k-1})$ and $\ell$ is the squared loss. This defines a recurring sequence in both population and over-parameterized regimes. In the population case, the sequence converges in parameter space to a fixed vector $\theta^*_{\text{pop}}$ (Section 4), while in the over-parameterized case the vector varies at each iteration but the excess risk still converges deterministically (Section 5).

We evaluate the test risk when the final model is deployed on the untouched distribution $\mathcal{D}(\theta = 0)$. This choice ensures that the final model is not evaluated on shifted distributions, and it is particularly relevant for long-term fairness, as it prevents bias amplification over time (Ensign et al., 2018;

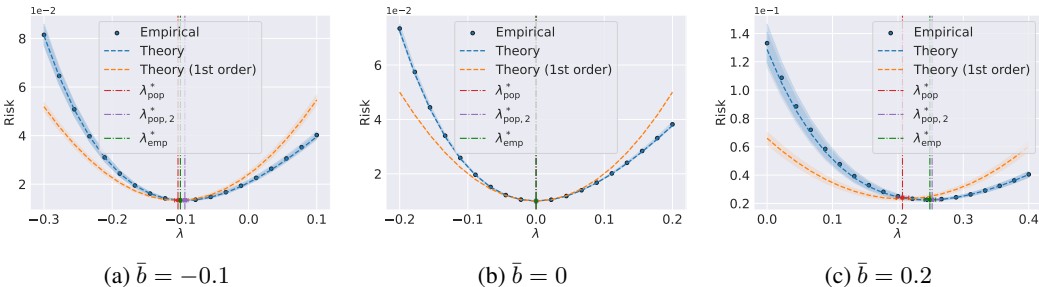

(a) $\bar{b} = -0.1$         (b) $\bar{b} = 0$         (c) $\bar{b} = 0.2$

Figure 1: Excess risk at the performative fixed point $\theta^\infty$ in (6), as a function of ridge regularization $\lambda$, for $d = 100$, $\Sigma = I_p$, entries of $b$ uniform in $\left[\min\{0, 2\bar{b}\}, \max\{0, 2\bar{b}\}\right]$, $c = 0$ and $\sigma = 0.1$. Empirical values (blue dots) are computed from 20 i.i.d. trials on $a$ and 5 i.i.d. trials on $b$, with error band at 1 standard deviation. Theoretical predictions (blue dashed curves) are from (7) and match perfectly empirical values. First-order approximations (orange dashed curves) are given by $\widetilde{\mathcal{R}}_{\mathrm{pop}}(D, \lambda, \Sigma)$ in (8) and still provide a good match when $\lambda$ is near-optimal. The green vertical line is the optimal regularization obtained by numerically optimizing the excess risk of $\theta^\infty$ ($\lambda^*_{\mathrm{emp}}$), the red one is the first-order approximation ($\lambda^*_{\mathrm{pop}}$ from (11)) and the violet one the second-order approximation ($\lambda^*_{\mathrm{pop},2}$ minimizing (12)).

Taori & Hashimoto, 2023) or steering the distribution toward undesirable regimes that decrease the risk by collapsing the data distribution to make it easier to predict (e.g., reducing entropy or producing a single possible label in a classification task) (Tsoy et al., 2025; Demirel et al., 2024). This choice also enables testing whether regularization increases reliance on spurious features (Bombari & Mondelli, 2025), which one wants to avoid. Furthermore, note that for the population setting, the in-distribution excess risk is always zero. For completeness, we also present results on $\mathcal{D}(\theta)$ for the proportional setting at the end of Section 5, deferring their formal statements and proofs to Appendix D. We thus aim to minimize the following excess risk:

$$\mathcal{R}(\Sigma, \theta, \theta^*_{\mathrm{pop}}) := \mathbb{E}_{\mathcal{D}(\theta=0)}\left[(y - x^\top \theta)^2]\right] - \sigma^2 = \|\Sigma^{1/2}(\theta - \theta^*_{\mathrm{pop}})\|_2^2, \tag{4}$$

where we have subtracted the Bayes risk $\sigma^2$. In Section 4, we analyze this risk in the population setting, where enough data is available to exactly recover the parameter vector. Here, the optimal solution is $\theta^*_{\mathrm{pop}}$, as suggested by the notation, and testing in-distribution instead of on $\mathcal{D}(\theta = 0)$ trivially yields zero risk, which also motivates the choice of reporting the excess risk on the untouched distribution. In Section 5, we then focus on the over-parameterized regime where $p > n$.

## 4 ANALYSIS IN THE POPULATION SETTING

In this section, we tackle the population regime where there are enough samples from $\mathcal{D}(\theta_k)$ at each deployment to compute exactly the next regressor, as would typically happen in a low-dimensional setting. The sequence $(\theta_k)_k$ is thus deterministically defined by

$$\theta_k = (\Sigma + \lambda I_p)^{-1}\mathbb{E}_{(x,y)\sim\mathcal{D}(\theta_{k-1})}[xy] = (\Sigma + \lambda I_p)^{-1}(\Sigma\theta^*_{\mathrm{pop}} + \Sigma D\theta_{k-1}), \tag{5}$$

where in the second equality we plug back the definition of the current data distribution in (1).

**Excess risk at the performative fixed point.** By unrolling (5), for any arbitrary possibly non-diagonal matrix $D$, we have that the sequence $(\theta_k)_k$ converges at an exponential rate to the fixed point

$$\theta^\infty = (I_p + \lambda\Sigma^{-1} - D)^{-1}\theta^*_{\mathrm{pop}}. \tag{6}$$

The formal statement, including an explicit convergence rate, is deferred to Lemma 5 in Appendix A. By inserting (6) into (4) and taking the expectation with respect to $\theta^*_{\mathrm{pop}}$, we have

$$\mathbb{E}_{\theta^*_{\mathrm{pop}}}\mathcal{R}(\Sigma, \theta^\infty, \theta^*_{\mathrm{pop}}) = \mathbb{E}_{\theta^*_{\mathrm{pop}}}[(\theta^*_{\mathrm{pop}})^\top A^\top \Sigma A\theta^*_{\mathrm{pop}}] = \frac{1}{d}\mathrm{Tr}\left[(A^\top\Sigma A)_1\right], \tag{7}$$

where we define $A := (\Sigma + \lambda I_p - \Sigma D)^{-1}\Sigma - I_p$, use $(\theta^*_{\mathrm{pop}})^\top = (a^\top, 0)$ with $a$ having zero mean and covariance $I_d/d$ and, given a $p \times p$ matrix $M$, denote by $(M)_1$ its top-left $d \times d$ block. This leads to the following approximation for the excess risk, also proved in Appendix A.

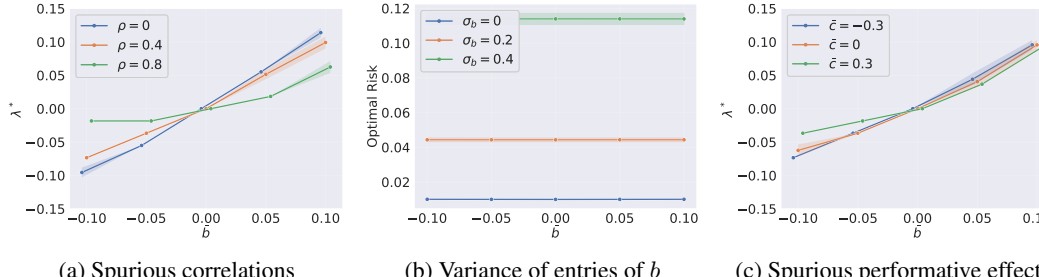

(a) Spurious correlations      (b) Variance of entries of $b$      (c) Spurious performative effect

Figure 2: Optimal regularization and risk for the performative fixed point $\theta^\infty$ in (6), with $d = 100$, $\Sigma_1 = \Sigma_2 = I_d, \Sigma_{12} = \rho I_d$. Values are computed from 20 i.i.d. trials on $a$ and 5 i.i.d. trials on $b$, with error band at 1 standard deviation. (a) Optimal regularization as a function of $\bar{b}$ for $\rho \in \{0, 0.4, 0.8\}$. The entries of $b$ are uniform in $\left[\min\{0, 2\bar{b}\}, \max\{0, 2\bar{b}\}\right]$ and $c = 0$. (b) Optimal risk as a function of $\bar{b}$. Different curves correspond to different variances $\sigma_b^2$ of the entries of $b$, which are uniform in $\left[\bar{b} - \sigma_b\sqrt{3}, \bar{b} + \sigma_b\sqrt{3}\right]$ for $\sigma_b \in \{0, 0.2, 0.4\}$. We pick $c = 0$ and $\rho = 0$. We note that, when $\rho = 0$, $\mathcal{R}^*_{\text{pop}}(D, \Sigma)$ equals the empirical variance of the entries of $b$ and, as such, it does not depend on $\bar{b}$. (c) Optimal regularization as a function of $\bar{b}$ for $\bar{c} \in \{-0.3, 0, 0.3\}$. The entries of $b$ are uniform in $\left[\min\{0, 2\bar{b}\}, \max\{0, 2\bar{b}\}\right]$, the entries of $c$ are uniform in $\left[\min\{0, 2\bar{c}\}, \max\{0, 2\bar{c}\}\right]$, and $\rho = 0.5$.

**Theorem 1** (Excess risk – population)**.** *Let $F = D - \lambda\Sigma^{-1}$. Then, we have*

$$\mathbb{E}_{\theta^*_{\text{pop}}}\mathcal{R}(\Sigma, \theta^\infty, \theta^*_{\text{pop}}) = \widetilde{\mathcal{R}}_{\text{pop}}(D, \lambda, \Sigma) + O(\|F\|^2_{\text{op}}),$$

$$\widetilde{\mathcal{R}}_{\text{pop}}(D, \lambda, \Sigma) := \frac{1}{d}\operatorname{Tr}[\operatorname{diag}(b^2)\Sigma_1] - 2\lambda\bar{b} + \frac{1}{d}\lambda^2\operatorname{Tr}(S_1), \tag{8}$$

*where $\|\cdot\|_{\text{op}}$ denotes the operator norm, $\bar{b} := \frac{1}{d}\operatorname{Tr}[\operatorname{diag}(b)] = \frac{1}{d}\sum_{i=1}^d b_i$, $b^2 := [b_1^2, \ldots, b_d^2] \in \mathbb{R}^d$ and $S_1 = (\Sigma_1 - \Sigma_{12}\Sigma_2^{-1}\Sigma_{21})^{-1}$ denotes the Schur complement of $\Sigma$.*

We note that the matrix $F = D - \lambda\Sigma^{-1}$ naturally appears in the computation, and an application of Weyl's inequality (see Lemma 6 in Appendix A) gives

$$\|F\|_{\text{op}} \le \max\left(\left|\max_{1 \le i \le d}\{b_i, c_i\} - \frac{\lambda}{\|\Sigma\|_{\text{op}}}\right|, \left|\min_{1 \le i \le d}\{b_i, c_i\} - \frac{\lambda}{\lambda_{\min}(\Sigma)}\right|\right), \tag{9}$$

where $\lambda_{\min}(\Sigma)$ is the smallest eigenvalue of $\Sigma$. From (9), we see that the approximation $O(\|F\|^2_{\text{op}})$ is tighter than $O(\max(\|b\|_\infty, \|c\|_\infty, \lambda)^2)$, since $b$ and $c$ can be partially canceled by $\lambda$. In fact, this cancellation occurs when $\lambda$ is near-optimal, resulting in an accurate approximation, see Figure 1.

**Optimal regularization and optimally regularized risk.** Leveraging the risk expression for small $F$ of Theorem 1, we next study the behavior of the optimal regularization and of the corresponding optimal risk. More formally, let us define

$$\lambda^*_{\text{pop}}(D, \Sigma) := \arg\min_{\lambda \in \mathbb{R}} \widetilde{\mathcal{R}}_{\text{pop}}(D, \lambda, \Sigma), \qquad \mathcal{R}^*_{\text{pop}}(D, \Sigma) := \min_{\lambda \in \mathbb{R}} \widetilde{\mathcal{R}}_{\text{pop}}(D, \lambda, \Sigma). \tag{10}$$

From (8), we note that $\widetilde{\mathcal{R}}_{\text{pop}}(D, \lambda, \Sigma)$ is quadratic in $\lambda$, so the minimization in (10) can be readily solved explicitly, leading to the expressions below.

**Corollary 2** (Optimal regularization – population)**.** *In the setting described above, we have*

$$\lambda^*_{\text{pop}}(D, \Sigma) = \frac{\bar{b}d}{\operatorname{Tr}(S_1)}, \qquad \mathcal{R}^*_{\text{pop}}(D, \Sigma) = \frac{1}{d}\operatorname{Tr}(\operatorname{diag}(b^2)\Sigma_1) - \frac{\bar{b}^2d}{\operatorname{Tr}(S_1)}. \tag{11}$$

These formulas call for several comments. First, the optimal regularization $\lambda^*_{\text{pop}}(D, \Sigma)$ is proportional to the strength of the performative effect $\bar{b}$, see Figure 2a (and also the location of the minima in Figure 1). The fact that $\lambda^*_{\text{pop}}(D, \Sigma)$ grows with $\bar{b}$ captures an effect common in practice: when performativity reinforces existing trends—corresponding to "rich-get-richer" phenomena, such as a feature becoming more important over successive deployments—the optimal regularizer increases and helps to limit this effect. Conversely, when the performative effect already mitigates the influence of some feature, the optimal solution calls for less regularization. In the population case, this

corresponds to negative regularization (see Figure 1a) which, although less common, has also been studied in the literature (Wu & Xu, 2020; Tsigler & Bartlett, 2023; Patil et al., 2024). Spurious correlations tend to reduce the optimal regularization, although their effect is mild, see Figure 2a.

Second, the optimal risk $\mathcal{R}^*_{\text{pop}}(D, \Sigma)$ is always positive and, thus, worse than in the non-performative scenario, where it is zero. More specifically, zero excess risk can only be reached with a non-zero $b$ if $\Sigma = I_p$ and $b$ is aligned with the all-1 vector, see the blue line in Figure 2b. Intuitively, as regularization impacts all features equally, it better compensates performativity in this uniform case. If the variance in the entries of $b$ grows, then $\mathcal{R}^*_{\text{pop}}(D, \Sigma)$ increases, see Figure 2b.

We finally note that $\tilde{\mathcal{R}}_{\text{pop}}(D, \lambda, \Sigma)$ does not depend on $c$ and, in fact, the effect of $c$ only becomes visible at higher order, as seen in this formula, proven in Appendix A:

$$
\begin{aligned}
\mathbb{E}_{\theta^*_{\text{pop}}} \mathcal{R}(\Sigma, \theta^\infty, \theta^*_{\text{pop}}) = \frac{1}{d} \Bigg( &-2\lambda^3 \operatorname{Tr}\big[(\Sigma^{-2})_1\big] + \lambda^2 \left(\operatorname{Tr}[S_1] + 6 \operatorname{Tr}\big[\operatorname{diag}(b) S_1\big]\right) \\
&- \lambda \Big( 2 \operatorname{Tr}\big[\operatorname{diag}(b)\Sigma_1 \operatorname{diag}(b) S_1\big] + 2 \operatorname{Tr}\big[\operatorname{diag}(b)\Sigma_{12} \operatorname{diag}(c) S_{21}\big] + 2d\bar{b} + 4 \operatorname{Tr}[\operatorname{diag}(b^2)]\Big) \\
&+ \operatorname{Tr}\big[\operatorname{diag}(b^2)\Sigma_1\big] + 2 \operatorname{Tr}\big[\operatorname{diag}(b^3)\Sigma_1\big] \Bigg) + O(\|F\|^4_{\text{op}}),
\end{aligned}
\tag{12}
$$

where $S^\top_{21} = -(\Sigma_1 - \Sigma_{12}\Sigma_2^{-1}\Sigma_{21})^{-1}\Sigma_{12}\Sigma_2^{-1}$ is the off-diagonal precision block. Figure 1 illustrates that the minimizer of this second-order approximation ($\lambda^*_{\text{pop},2}$) is close to the minimizer obtained numerically ($\lambda^*_{\text{emp}}$). While $c$ tends to steer the optimal regularizer in the opposite direction (less regularization in the case of a self-reinforcing performative effect), it does so only through the cross term $2 \operatorname{Tr}[\operatorname{diag}(b)\Sigma_{12} \operatorname{diag}(c) S_{21}]$, which also depends on $b$. Figure 2c shows that $c$ moves the optimal regularization in a direction opposite to its sign, but its effect remains rather limited.

## 5 ANALYSIS IN THE OVER-PARAMETERIZED SETTING

Next, we consider the case where $n, p$ are both large and scale proportionally, with $p/n = \kappa > 1$. All *constants* (e.g., $R, M$) are intended to be positive values independent of $n, p$. For mathematical convenience, we opt for a different normalization w.r.t. (3), and the estimator $\theta_k$ is given by

$$
\theta_k = \arg\min_{\theta \in \mathbb{R}^p} \left\{ \frac{1}{2p} \sum_{i=1}^n \ell\left(x_i^{(k-1)}, y_i^{(k-1)}; \theta\right) + \frac{\lambda}{2}\|\theta\|_2^2 \right\}.
\tag{13}
$$

Solving for $\theta_k$ yields

$$
\theta_k = \frac{1}{p}\left(\frac{1}{p}X^{(k-1)\top}X^{(k-1)} + \lambda I_p\right)^{-1} X^{(k-1)\top}y^{(k-1)},
\tag{14}
$$

where $X^{(k-1)} = [x_1^{(k-1)}, \ldots, x_n^{(k-1)}] \in \mathbb{R}^{n \times p}$ and $y^{(k-1)} = [y_1^{(k-1)}, \ldots, y_n^{(k-1)}] \in \mathbb{R}^n$. Note that, when $\theta_k$ is given by (14), the risk $\mathcal{R}(\Sigma, \theta_k, \theta^*_{\text{pop}})$ as defined in (4) is a random quantity since the data $\{X^{(\ell)}\}_{\ell=1}^{k-1}$ and the noise contained in the labels $\{y^{(\ell)}\}_{\ell=1}^{k-1}$ are random. This makes it challenging to characterize optimal ridge penalty and optimally-tuned risk. To address the challenge, we first establish a *deterministic* equivalent of the risk at the performative fixed point. We next optimize such deterministic equivalent and study the effect of performativity on the optimal regularization.

**Deterministic equivalent of the performative fixed point.** First, note that, if we regard the performative effect as small and aim at characterizing its effect on the fixed point up to the leading (first) order, it suffices to do two iterations of the recursion in (14). In fact, the labels $y^{(k-1)}$ are linear in $D\theta_{k-1}$, so we expect that, after two iterations, the performative fixed point is reached up to fluctuations of order $O(\|D\|^2_{\text{op}})$. This is still a random quantity, so we apply techniques from Han & Xu (2023) (and more precisely from Ildiz et al. (2025)) to derive a deterministic equivalent.

**Theorem 3** (Excess risk – over-parameterized). *Let Assumption 1 hold. Let $R > 0$ be a constant s.t. $\theta^*_{\text{pop}}, \theta_0 \in B_p(R)$. Assume that $\kappa, \sigma, \lambda \in (1/M, M)$ and $\|\Sigma\|_{\text{op}}, \|\Sigma^{-1}\|_{\text{op}} \le M$ for some constant $M > 1$. Then, there exists a constant $C = C(M, R)$ such that for any $\delta \in (0, 1/2)$, with probability at least $1 - Cpe^{-p\delta^4/C}$,*

$$
\left|\mathcal{R}(\Sigma, \theta_2, \theta^*_{\text{pop}}) - \mathcal{R}_{\text{eq}}\left(\Sigma, \theta^*_{\text{pop}}, D, \lambda\right)\right| \le \delta + O(\|D\|^2_{\text{op}}),
\tag{15}
$$

*where*

$$\mathcal{R}_{\mathrm{eq}}\left(\Sigma, \theta_{\mathrm{pop}}^*, D, \lambda\right) = \tau \langle \theta_{\mathrm{pop}}^*, (\Sigma + \tau I_p)^{-1} \left(\tau I_p - 2(\Sigma + \tau I_p)^{-1}\Sigma^2 D\right)\Sigma(\Sigma + \tau I_p)^{-1}\theta_{\mathrm{pop}}^*\rangle$$

$$+\kappa \operatorname{Tr}\left[\Sigma^2 \left(\Sigma + \tau I_p\right)^{-2}\right] \frac{\sigma^2 + \tau^2 \langle \theta_{\mathrm{pop}}^*, (\Sigma + \tau I_p)^{-1}\left(I_p + 2\left(\Sigma + \tau I_p\right)^{-1}\Sigma D\right)\Sigma\left(\Sigma + \tau I_p\right)^{-1}\theta_{\mathrm{pop}}^*\rangle}{p - \kappa \operatorname{Tr}\left[\Sigma^2\left(\Sigma + \tau I_p\right)^{-2}\right]},$$

$$(16)$$

*and $\tau$ is the unique solution of*

$$\kappa^{-1} - \frac{\lambda}{\tau} = \frac{1}{p}\operatorname{Tr}\left[(\Sigma + \tau I_p)^{-1}\Sigma\right]. \tag{17}$$

In words, Theorem 3 shows that the risk $\mathcal{R}(\Sigma, \theta_2, \theta_{\mathrm{pop}}^*)$ is well approximated by the quantity $\mathcal{R}_{\mathrm{eq}}\left(\Sigma, \theta_{\mathrm{pop}}^*, D, \lambda\right)$ defined in (16). We highlight that this quantity does not depend on the initialization $\theta_0$: up a fluctuation of order $O(\|D\|_{\mathrm{op}}^2)$, *the risk has reached a fixed point* after two iterations. While the data–and consequently $\mathcal{R}(\Sigma, \theta_2, \theta_{\mathrm{pop}}^*)$–are random, $\mathcal{R}_{\mathrm{eq}}\left(\Sigma, \theta_{\mathrm{pop}}^*, D, \lambda\right)$ provides a *deterministic equivalent* that depends only on the population covariance $\Sigma$, the ground-truth vector $\theta_{\mathrm{pop}}^*$, the matrix $D$ capturing the performative effect and the regularization $\lambda$.

The assumptions ($\theta_{\mathrm{pop}}^*, \theta_0 \in B_p(R)$, $\kappa, \sigma, \lambda \in (1/M, M)$, $\|\Sigma\|_{\mathrm{op}}$, $\|\Sigma^{-1}\|_{\mathrm{op}} \leq M$) are all standard in the related literature (Han & Xu, 2023; Ildiz et al., 2025). We could handle the ridgeless case $\lambda = 0$ in a similar way to Han & Xu (2023); Ildiz et al. (2025). However, this requires changing some details and we have opted to avoid the notation clutter, since our focus is on the effect of regularization. We note that the result of Theorem 3 holds for a general matrix $D$ with bounded operator norm—not necessarily a diagonal $D$ as in Assumption 1. The assumption on the features $x$ being Gaussian can be also relaxed at the cost of a more involved analysis. In fact, the results of Han & Xu (2023) (see Theorem 2.4 therein) hold for $\Sigma^{-1/2}x$ having independent, zero mean, unit variance and uniformly subgaussian entries. We elaborate on this in Appendix B.1. The proof of Theorem 3 is deferred to Appendix B, and we give a sketch below.

*Proof sketch.* Having fixed the initialization $\theta_0$, the only randomness in $\mathcal{R}(\Sigma, \theta_1, \theta_{\mathrm{pop}}^*)$ comes from $(X^{(0)}, y^{(0)})$. This corresponds to the setting in which one trains using the vector of regression coefficients $\theta_{\mathrm{pop}}^* + D\theta_0$ and then tests on $\theta_{\mathrm{pop}}^*$. Lemma 7 in Appendix B (which follows from Theorem 3 by Ildiz et al. (2025) and uses the non-asymptotic characterization of the minimum norm interpolator by Han & Xu (2023)) gives that, with high probability, $\mathcal{R}(\Sigma, \theta_1, \theta_{\mathrm{pop}}^*)$ is close to

$$\mathcal{R}_{\mathrm{eq}}^{(1)}\left(\Sigma, \theta_0, \theta_{\mathrm{pop}}^*\right) = \left\|(\Sigma + \tau I_p)^{-1}\Sigma(\theta_{\mathrm{pop}}^* + D\theta_0) - \theta_{\mathrm{pop}}^*\right\|_\Sigma^2$$

$$+\kappa \operatorname{Tr}\left[\Sigma^2\left(\Sigma + \tau I_p\right)^{-2}\right]\frac{\sigma^2 + \tau^2 \left\|(\Sigma + \tau I_p)^{-1}\left(\theta_{\mathrm{pop}}^* + D\theta_0\right)\right\|_\Sigma^2}{p - \kappa \operatorname{Tr}\left[\Sigma^2\left(\Sigma + \tau I_p\right)^{-2}\right]}, \tag{18}$$

where $\|x\|_\Sigma^2 := x^\top \Sigma x$. We note that the expression in (18) depends on $\theta_0$ and, in fact, it keeps depending on it even after neglecting terms of order $O(\|D\|_{\mathrm{op}}^2)$. This means that we have not yet reached the fixed point. Thus, we turn our attention to the risk after two iterations and, by iterating twice the strategy of Lemma 7, we show that, with high probability, $\mathcal{R}(\Sigma, \theta_2, \theta_{\mathrm{pop}}^*)$ is close to $\mathcal{R}_{\mathrm{eq}}^{(2)}\left(\Sigma, \theta_0, \theta_{\mathrm{pop}}^*\right)$, whose expression is given in Lemma 8 in Appendix B. Finally, by neglecting terms of order $O(\|D_{\mathrm{op}}\|^2)$ in $\mathcal{R}_{\mathrm{eq}}^{(2)}\left(\Sigma, \theta_0, \theta_{\mathrm{pop}}^*\right)$, this quantity equals $\mathcal{R}_{\mathrm{eq}}\left(\Sigma, \theta_{\mathrm{pop}}^*, D, \lambda\right)$ in (16), and the desired result follows. $\square$

**Optimal regularization and optimally regularized risk.** Leveraging the characterization of Theorem 3, we optimize the ridge regularization. We focus on the case $\Sigma = \begin{bmatrix} I_d & \rho I_d \\ \rho I_d & I_d \end{bmatrix}$ for small $\rho$ and require $D$ to be diagonal as in Assumption 1. While simplified, this setting captures the performative effect of both predictive and spurious features, which are mixed via the covariance matrix $\Sigma$, leading to an interesting phenomenology. Lemma 12 in Appendix C computes $\mathbb{E}_{\theta_{\mathrm{pop}}^*}\mathcal{R}_{\mathrm{eq}}\left(\Sigma, \theta_{\mathrm{pop}}^*, D, \lambda\right)$,

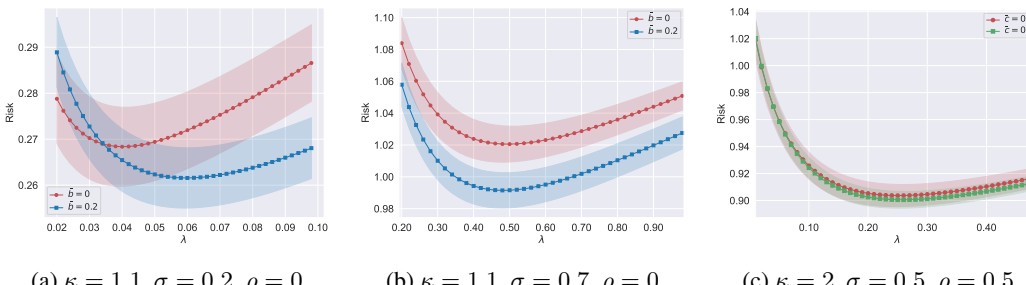

(a) $\kappa = 1.1$, $\sigma = 0.2$, $\rho = 0$     (b) $\kappa = 1.1$, $\sigma = 0.7$, $\rho = 0$     (c) $\kappa = 2$, $\sigma = 0.5$, $\rho = 0.5$

Figure 3: Excess risk as a function of ridge regularization $\lambda$ with Gaussian data, for $n = 4000$, $\Sigma_1 = \Sigma_2 = I_d$, $\Sigma_{12} = \rho I_d$, entries of $b$ equal to $\bar{b}$, and entries of $c$ equal to $\bar{c}$. Values are computed from 20 i.i.d. trials, with error band at 1 standard deviation. We perform 5 steps of RRM to approximate the fixed point, as in the simulation setup of Section 6. (a) In the low-noise regime ($\sigma = 0.2$), taking $\bar{b} = 0.2$ instead of $\bar{b} = 0$ *increases the optimal regularization* and *reduces the optimal risk*. We set $\bar{c} = 0$ to emphasize the dependence on $\bar{b}$. (b) In the large-noise regime ($\sigma = 0.7$), taking $\bar{b} = 0.2$ instead of $\bar{b} = 0$ *reduces both optimal regularization and optimal risk*. As in (a), we set $\bar{c} = 0$. (c) Taking $\bar{c} = 0.2$ instead of $\bar{c} = 0$ *reduces the optimal risk*, although the the impact of $\bar{c}$ is less pronounced. We set $\bar{b} = 0$ to emphasize the dependence on $\bar{c}$.

as well as the following expansion in $\rho$:

$$\mathbb{E}_{\theta^*_{\text{pop}}} \mathcal{R}_{\text{eq}}\left(\Sigma, \theta^*_{\text{pop}}, D, \lambda\right) = \widetilde{\mathcal{R}}(D, \lambda, \rho) + O(\bar{b}\rho^2 + \rho^4),$$
$$\widetilde{\mathcal{R}}(D, \lambda, \rho) := \mathcal{R}_0(\lambda, \rho) + \bar{b} A_1(\lambda) + \bar{c}\rho^2 A_2(\lambda),$$
(19)

with $\bar{b} = \text{Tr}[\text{diag}(b)]/d$, $\bar{c} = \text{Tr}[\text{diag}(c)]/d$. Explicit expressions for $\mathcal{R}_0(\lambda, \rho)$, $A_1(\lambda)$ and $A_2(\lambda)$ are given in (56) in Appendix C. We note that $\mathbb{E}_{\theta^*_{\text{pop}}} \mathcal{R}_{\text{eq}}\left(\Sigma, \theta^*_{\text{pop}}, D, \lambda\right)$ is even in $\rho$, hence the odd powers of $\rho$ are absent from (19). Now, we define optimal regularization and risk as

$$\lambda^*_{\text{eq}}(D, \rho) := \arg\min_{\lambda \geq 0} \widetilde{\mathcal{R}}(D, \lambda, \rho), \qquad \mathcal{R}^*_{\text{eq}}(D, \rho) := \min_{\lambda \geq 0} \widetilde{\mathcal{R}}(D, \lambda, \rho).$$
(20)

Our goal is to characterize the performative effect on $\lambda^*_{\text{eq}}(D, \lambda, \rho)$, $\mathcal{R}^*_{\text{eq}}(D, \lambda, \rho)$ and, to do so, we compare these quantities to their values when $D = 0$, defined as

$$\lambda^*_{\text{eq}, D=0}(\rho) := \arg\min_{\lambda \geq 0} \mathcal{R}_0(\lambda, \rho), \qquad \mathcal{R}^*_{\text{eq}}(\rho) = \min_{\lambda \geq 0} \mathcal{R}_0(\lambda, \rho).$$
(21)

This is formalized by the result below whose proof is deferred to Appendix C.

**Theorem 4** (Optimal regularization – over-parameterized). *In the setting described above, we have*

$$\lambda^*_{\text{eq}}(D, \rho) = \lambda^*_{\text{eq}, D=0}(\rho) + \bar{b}(B_1(\sigma, \kappa) + O(\rho^2)) + \bar{c}\rho^2(C_1(\sigma, \kappa) + O(\rho^2)) + O(\bar{b}^2 + \bar{c}^2), \quad (22)$$
$$\mathcal{R}^*_{\text{eq}}(D, \rho) = \mathcal{R}^*_{\text{eq}}(\rho) + \bar{b}(B_2(\sigma, \kappa) + O(\rho^2)) + \bar{c}\rho^2(C_2(\sigma, \kappa) + O(\rho^2)) + O(\bar{b}^2 + \bar{c}^2), \quad (23)$$

*where the functions $B_1(\sigma, \kappa), B_2(\sigma, \kappa), C_1(\sigma, \kappa), C_2(\sigma, \kappa)$ depend only on $\sigma, \kappa$ and they are explicitly given in* (61)-(63). *Furthermore, these functions satisfy*

$$B_1(\sigma, \kappa) \geq 0 \quad \text{for } 0 \leq \sigma \leq \sigma_{B_1}(\kappa), \ \kappa > 1, \qquad B_1(\sigma, \kappa) \leq 0 \quad \text{for } \sigma > \sigma_{B_1}(\kappa), \ \kappa > 1, \quad (24)$$
$$C_1(\kappa, \sigma) \leq 0 \quad \text{for } \sigma \geq 0, \kappa \geq 2, \quad (25)$$
$$B_2(\kappa, \sigma) \leq 0 \quad \text{for } \sigma \geq 0, \kappa > 1, \quad (26)$$
$$C_2(\kappa, \sigma) \leq 0 \quad \text{for } \sigma \geq 0, \kappa > 1, \quad (27)$$

*with* $\sigma^2_{B_1}(\kappa) = 1/2 - 7\kappa^{-1}/18 + O(\kappa^{-2})$.

In words, (22) gives a quantitative comparison between optimal regularization with performative effect ($\lambda^*_{\text{eq}}(D, \rho)$) and without it ($\lambda^*_{\text{eq}, D=0}(\rho)$). Similarly, (23) compares optimally-regularized risks $\mathcal{R}^*_{\text{eq}}(D, \rho)$ and $\mathcal{R}^*_{\text{eq}}(\rho)$ respectively with and without performativity. The study of the signs of the auxiliary functions $B_1(\sigma, \kappa), B_2(\sigma, \kappa), C_1(\sigma, \kappa), C_2(\sigma, \kappa)$ leads to the considerations below:

- Equation (24) implies that *(i)* if the noise variance $\sigma^2$ is small, then the optimal regularization moves in the same direction as the performative effect on the predictive features; and *(ii)* if the noise variance is large, the effect is reversed and the optimal regularization moves in the opposite direction to the performative effect. This is illustrated in Figures 3a and 3b.

- Equation (25) implies that, when $\kappa \geq 2$, the optimal regularization moves in the opposite direction to the performative effect on the spurious features. This effect is however significantly attenuated by the factor $\rho^2$ multiplying $\bar{c}$ in (22) and, as such, it is hardly noticeable both with Gaussian data (as considered in this section) and in real-world settings (as considered in Section 6).

- Equation (26) implies that, when performativity reinforces existing trends ($\bar{b} > 0$), the optimally-regularized risk improves in the presence of a performative effect on the predictive features. This occurs regardless of the size of the noise variance, and it is illustrated in Figures 3a and 3b. Instead, when performativity dampens existing trends ($\bar{b} < 0$), the effect is reversed and the optimal risk worsens.

- Finally, Equation (27) implies that the dependence of the optimally-regularized risk on the performative effect on the spurious features is analogous: the optimal risk decreases when $\bar{c} > 0$, and increases when $\bar{c} < 0$. However, as for $\lambda_{\mathrm{eq}}^*(D, \rho)$, the impact of performativity on $\mathcal{R}_{\mathrm{eq}}^*(D, \rho)$ is less pronounced for the spurious features, due to the factor $\rho^2$ multiplying $\bar{c}$ in (23). This is illustrated in Figure 3c.

**Evaluating the model on the shifted distribution.** We finally provide an analysis in the over-parameterized setting of the model tested on the shifted distribution (i.e., on the same distribution used to train it). More precisely, Theorem 19 is the equivalent of Theorem 3, and it provides a deterministic equivalent of the performative fixed point. Lemma 20 specializes the risk expression from Theorem 19 to the case in which the covariance has the form $\Sigma = \begin{bmatrix} I_d & \rho I_d \\ \rho I_d & I_d \end{bmatrix}$, and it is the equivalent of Lemma 12. Next, we provide expressions for the optimal $\tau$ in Lemma 21, for the optimal regularization parameter in (75)-(76), and for the optimal risk in Corollary 22. Lemma 23 shows that the optimal regularization moves in the same direction as the performative effect on the predictive features, as it is the case for testing over $\mathcal{D}(\theta = 0)$ provided that the noise variance is small. Finally, Lemmas 24 and 25 show that the optimally regularized risk worsens in the presence of a performative effect (either on the predictive or on the spurious features) that reinforces existing trends ($\bar{b}, \bar{c} > 0$). This is in contrast with the behavior of the optimally regularized risk tested over $\mathcal{D}(\theta = 0)$ which instead decreases when either $\bar{b} > 0$ or $\bar{c} > 0$. All these results are formally stated and proved in Appendix D.

## 6 NUMERICAL EXPERIMENTS

In this section, we test the effect of regularization and performative shifts on real data. Since no dataset currently provides a real performative shift, it must be encoded synthetically. In practice, we take a real-world dataset, randomly split the samples across time steps, train a model on one split, compute the parameter $\theta$, and then shift the samples of the next split according to the theoretical model. This methodology follows previous work (Perdomo et al., 2020; Hardt & Mendler-Dünner, 2023; Zezulka & Genin, 2023). These experiments allow us to test whether the theory remains predictive when *(i)* the data is non-Gaussian and $\theta_{\mathrm{pop}}^*$ is fixed by the task, and *(ii)* the true relationship between the feature and the target is likely to not be linear.

We consider two datasets. First, we use the Housing dataset,[1] where the goal is to predict house prices from housing features and local demographics. We follow the methodology of Cyffers et al. (2024) to choose performative features. The dataset has 8 features and 20,640 datapoints, which we split into five folds: four for training and one for test. Four training steps suffice experimentally to reach the fixed point, which is consistent with the theory, where the first-order effect stabilizes after only two iterations. Second, we use the Law School Admission Council (LSAC) dataset,[2] where the default task is to predict bar passage from demographic features and previous grades. We change the target to GPA to maintain a regression task, and randomly choose which features are affected by performativity. After dropping redundant columns or those too correlated with GPA, the dataset has 22 features and 20,427 samples, which again we split in five folds. We report detailed pre-processing and parameters in Appendix E.

---

[1] https://www.openml.org/d/823

[2] https://storage.googleapis.com/lawschool_dataset/bar_pass_prediction.csv

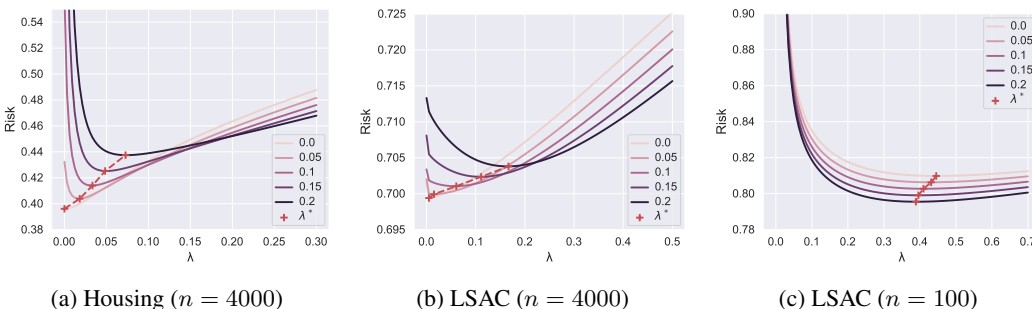

(a) Housing ($n = 4000$)  (b) LSAC ($n = 4000$)  (c) LSAC ($n = 100$)

Figure 4: Excess risk as a function of ridge regularization $\lambda$ in real-world datasets (Housing, LSAC). Different curves (in different colors) correspond to different values of $\bar{b} \in \{0, 0.05, 0.1, 0.15, 0.2\}$, and we connect with a red dashed line the optima of the risk for various choices of $\bar{b}$. The plots in (a)-(b) use $n = 4000$ data points at each training step, which corresponds to the population setting ($n \gg d$); the plot in (c) uses $n = 100$, a value closer to the number of features $d = 22$.

When $n = 4000$, both for the Housing (Figure 4a) and the LSAC (Figure 4b) dataset, we note that *(i)* the optimal regularizer increases proportionally to $\bar{b}$, and *(ii)* the optimally-regularized risk becomes worse as $\bar{b}$ grows. This can be attributed to the fact that $n \gg d$, and it is consistent with our theoretical results in the population setting (Corollary 2). In contrast, when training with very few samples on LSAC (Figure 4c), the behavior of the regularized risk follows the predictions of Theorem 3 for the proportional setting in the large-noise regime: as $\bar{b}$ grows, the optimal regularizer gets smaller and the risk improves. Note that, even if Figures 4b and 4c consider the same dataset, the ranges of the excess risk and the regularizer are not the same due to the different sample sizes. We did not find numerical evidence for the role of $c$, suggesting that its effect may be dominated by data noise, consistent with our theoretical findings on the limited impact of spurious features.

## 7 CONCLUSIONS

In this work, we demonstrate that regularization and performative effect are strongly related, as one can partially cancel out the other. In the population regime, the excess risk is worsened by performative effects. However, optimal ridge regularization mitigates this issue, especially when the data is isotropic and the entries of the vector modeling performativity have little variability. In the proportional regime, we provide a deterministic equivalent of the performative fixed point for random data. This in turn unveils a remarkable phenomenology: in contrast with the population setting, the optimal risk improves when performativity reinforces existing trends; furthermore, the optimal regularization follows the direction of the performative effect on the predictive features when the noise is small, while it goes in the opposite direction when the noise is large. Although the theoretical results focus on Gaussian features and a linear target model, our experiments on real-world data follow the theoretical predictions, suggesting their generality. Overall, these findings indicate that regularization could help in a wider range of scenarios, which we leave for future work. Beyond studying more complex data or models, interesting directions include the impact of other forms of regularization, such as early stopping or pruning, to mitigate performative effects.

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

## A    ADDITIONAL PROOFS FOR SECTION 4

This appendix contains the missing proofs for Section 4. We start with the convergence at exponential rate to the fixed point $\theta^\infty$ in (6). Then, we prove Theorem 1 giving the first-order approximation of the risk, as well as the expression in (12) giving the higher-order approximation of the risk. Finally, we prove the upper bound on $\|F\|_{\mathrm{op}}$ in (9).

**Lemma 5.** *The sequence $(\theta_k)_k$ converges to the fixed point*

$$\theta^\infty = (\Sigma + \lambda I_p - M)^{-1}\Sigma\theta^*_{\mathrm{pop}}.$$

*Moreover, for any $\varepsilon \in (0, 1)$, if we start at $\theta_0 = 0$, after at most*

$$k_\varepsilon = \left\lceil \frac{\ln(1/\varepsilon)}{\ln\left(1/\left(\frac{\|\Sigma\|_{\mathrm{op}}}{\|\Sigma\|_{\mathrm{op}}+\lambda}\max\{\|b\|_\infty, \|c\|_\infty\}\right)\right)} \right\rceil$$

*iterations, the relative error $\frac{\|\theta^{k_\varepsilon}-\theta^\infty\|_2}{\|\theta^\infty\|_2}$ is smaller than $\varepsilon$.*

*Proof.* Denoting $T = (\Sigma + \lambda I_p)^{-1}\Sigma D$, the recurrence relation is

$$\theta^k = T\theta^{k-1} + (\Sigma + \lambda I_p)^{-1}\Sigma\theta^*_{\mathrm{pop}}.$$

When going to the limit, $\sum_i T^i \to (I_p - T)^{-1}$. The convergence requires the matrix $T$ to have smaller eigenvalues than one, which is guaranteed by $\|b\|_\infty$ and $\|c\|_\infty$ being smaller than 1. Thus, we have

$$\theta^\infty = (I_p - T)^{-1}(\Sigma + \lambda I_p)^{-1}\Sigma\theta^*_{\mathrm{pop}}.$$

Noticing that $I_p = (\Sigma + \lambda I_p)^{-1}(\Sigma + \lambda I_p)$ and using the definition of $T$ gives the expression of $\theta_\infty$.

Let $e_k = \theta^k - \theta^\infty$. Using $\theta^\infty = T\theta^\infty + (\Sigma + \lambda I_p)^{-1}\Sigma\,\theta^*_{\mathrm{pop}}$, we have

$$e_k = \theta^k - \theta^\infty = T\theta^{k-1} + (\Sigma + \lambda I_p)^{-1}\Sigma\theta^*_{\mathrm{pop}} - \theta^\infty = T(\theta^{k-1} - \theta^\infty) = Te_{k-1}.$$

Thus,

$$e_k = T^k e_0 = -T^k\theta^\infty \implies \|e_k\|_2 \le \|T\|^k_{\mathrm{op}}\|\theta^\infty\|_2 \implies \frac{\|e_k\|_2}{\|\theta^\infty\|_2} = \frac{\|\theta^k - \theta^\infty\|_2}{\|\theta^\infty\|_2} \le \|T\|^k_{\mathrm{op}}.$$

Consequently, $\|T\|^k \le \varepsilon$ suffices, i.e., $k \ge \ln(1/\varepsilon)/\ln(1/\|T\|_{\mathrm{op}})$. We finally note that

$$\|T\|_{\mathrm{op}} \le \|(\Sigma + \lambda I_p)^{-1}\Sigma\|_{\mathrm{op}}\|D\|_{\mathrm{op}} = \frac{\|\Sigma\|_{\mathrm{op}}}{\|\Sigma\|_{\mathrm{op}} + \lambda}\max\{\|b\|_\infty, \|c\|_\infty\}.$$

Combining the last two inequalities gives the wanted convergence rate. $\qquad\square$

*Proof of Theorem 1 and of the higher-order approximation in (12).* We start by computing the Taylor expansion:

$$A = (\Sigma + \lambda I_p - \Sigma D)^{-1}\Sigma - I_p = (I_p - (D - \lambda\Sigma^{-1}))^{-1} - I_p = \sum_{i=1}^\infty (D - \lambda\Sigma^{-1})^i = \sum_{i=1}^\infty F^i.$$

Let us define $A^{(k)} = \sum_{i=1}^k (D - \lambda\Sigma^{-1})^i$. For the two first orders, we have:

$$A^{(1)\,\top}\Sigma A^{(1)} = (D - \lambda\Sigma^{-1})\Sigma(D - \lambda\Sigma^{-1}) = D\Sigma D - 2\lambda D + \lambda^2\Sigma^{-1}.$$

This is independent of $c$ and gives the simple formula

$$R^{(1)}(\lambda) = \frac{1}{d}\mathrm{Tr}(\mathrm{diag}(b^2)\Sigma_1) - 2\lambda\bar{b} + \frac{1}{d}\lambda^2\,\mathrm{Tr}(S_1),$$

where $\bar{b} := \frac{1}{d}\mathrm{Tr}[\mathrm{diag}(b)] = \frac{1}{d}\sum_{i=1}^d b_i$, $b^2 := [b_1^2, \ldots, b_d^2] \in \mathbb{R}^d$ and $S_1 = (\Sigma_1 - \Sigma_{12}\Sigma_2^{-1}\Sigma_{21})^{-1}$ denotes the Schur complement of $\Sigma$. We go further in the expansion to recover (12):

$$
A^{(2)\top}\Sigma A^{(2)} = A^{(1)\top}\Sigma A^{(1)} + A^{(1)\top}\Sigma\left(A^{(2)} - A^{(1)}\right) + \left(A^{(2)} - A^{(1)}\right)^\top \Sigma A^{(1)}
$$

$$
+ \underbrace{\left(A^{(2)} - A^{(1)}\right)^\top \Sigma \left(A^{(2)} - A^{(1)}\right)}_{O\left(\|F\|_{\mathrm{op}}^4\right),\ \mathrm{discarded}}
$$

$$
= D\Sigma D + D^2\Sigma D + D\Sigma D^2 - \lambda\left[D\Sigma D\Sigma^{-1} + \Sigma^{-1}D\Sigma D + 2D + 4D^2\right]
$$

$$
+ \lambda^2\left[\Sigma^{-1} + 3(\Sigma^{-1}D + D\Sigma^{-1})\right] - 2\lambda^3\Sigma^{-2} + O\left(\|F\|_{\mathrm{op}}^4\right).
$$

The final formula results from taking the trace of the first block. We write the matrix product block per block to prove that

$$
\mathrm{Tr}\left[(D\Sigma D\Sigma^{-1} + \Sigma^{-1}D\Sigma D)_1\right] = 2\,\mathrm{Tr}\left[\mathrm{diag}(b)\Sigma_1\,\mathrm{diag}(b)S_1\right] + 2\,\mathrm{Tr}\left[\mathrm{diag}(b)\Sigma_{12}\,\mathrm{diag}(c)S_{21}\right],
$$

where $S_{21}^\top = -(\Sigma_1 - \Sigma_{12}\Sigma_2^{-1}\Sigma_{21})^{-1}\Sigma_{12}\Sigma_2^{-1}$. This concludes the proof. $\qquad\square$

**Lemma 6.** *Let $F = D - \lambda\Sigma^{-1}$. Then, we have that*

$$
\|F\|_{\mathrm{op}} \le \max\left(\left|\max_{1\le i\le d}\{b_i, c_i\} - \frac{\lambda}{\lambda_{\max}(\Sigma)}\right|, \left|\min_{1\le i\le d}\{b_i, c_i\} - \frac{\lambda}{\lambda_{\min}(\Sigma)}\right|\right). \tag{28}
$$

*Proof.* By Weyl's inequalities for Hermitian matrices,

$$
\lambda_{\max}(F) \le \lambda_{\max}(D) - \lambda\lambda_{\min}(\Sigma^{-1}) = \max_{1\le i\le d}\{b_i, c_i\} - \frac{\lambda}{\lambda_{\max}(\Sigma)},
$$

$$
\lambda_{\min}(F) \ge \lambda_{\min}(D) - \lambda\lambda_{\max}(\Sigma^{-1}) = \min_{1\le i\le d}\{b_i, c_i\} - \frac{\lambda}{\lambda_{\min}(\Sigma)}.
$$

Therefore, we have

$$
\|F\|_{\mathrm{op}} = \max\left\{|\lambda_{\max}(F)|, |\lambda_{\min}(F)|\right\}
$$

$$
\le \max\left(\left|\max_{1\le i\le d}\{b_i, c_i\} - \frac{\lambda}{\lambda_{\max}(\Sigma)}\right|, \left|\min_{1\le i\le d}\{b_i, c_i\} - \frac{\lambda}{\lambda_{\min}(\Sigma)}\right|\right),
$$

and the equality happens if and only if and only if $\Sigma$ and $D$ are simultaneously diagonalizable. $\quad\square$

We finally note that (28) coincides with (9) since $\lambda_{\max}(\Sigma) = \|\Sigma\|_{\mathrm{op}}$, due to $\Sigma$ being a covariance matrix and, hence, positive semidefinite.

## B  PROOF OF THEOREM 3

**Deterministic equivalent for $\mathcal{R}_1(\Sigma, \theta_1, \theta_{\mathrm{pop}}^*)$.** Let $\mathcal{R}_k(\Sigma, \theta_k, \theta_{\mathrm{pop}}^*)$ be the excess risk of the estimator $\theta_k$ given by (14), i.e.,

$$
\mathcal{R}_k(\Sigma, \theta_k, \theta_{\mathrm{pop}}^*) = \left\|\theta_k - \theta_{\mathrm{pop}}^*\right\|_\Sigma^2.
$$

Having fixed the initialization $\theta_0$, the only randomness in $\mathcal{R}_1(\Sigma, \theta_1, \theta_{\mathrm{pop}}^*)$ comes from $(X^{(0)}, y^{(0)})$. This corresponds to the setting in which one trains from the (deterministic) vector of regression coefficients $\theta_{\mathrm{pop}}^* + D\theta_0$. The following lemma gives a deterministic equivalent for $\mathcal{R}_1(\Sigma, \theta_1, \theta_{\mathrm{pop}}^*)$, conditional on $\theta_0$.

**Lemma 7.** *Let Assumption 1 hold. Let $R > 0$ be a constant such that $\theta_{\mathrm{pop}}^*, \theta_0 \in B_p(R)$. Assume that $\kappa, \sigma, \lambda \in (1/M, M)$ and $\|\Sigma\|_{\mathrm{op}}, \|\Sigma^{-1}\|_{\mathrm{op}} \le M$ for some constant $M > 1$. Then, there exists a constant $C = C(M, R)$ such that, for any $\delta \in (0, 1/2]$, the following holds*

$$
\sup_{\theta_{\mathrm{pop}}^*, \theta_0 \in B_p(R)} \mathrm{Pr}\left(\left|\mathcal{R}_1(\Sigma, \theta_1, \theta_{\mathrm{pop}}^*) - \mathcal{R}_{\mathrm{eq}}^{(1)}\left(\Sigma, \theta_0, \theta_{\mathrm{pop}}^*\right)\right| \ge \delta\right) \le Cpe^{-p\delta^4/C}, \tag{29}
$$

*with probability at least $1 - Cpe^{-p\delta^4/C}$, where*

$$\mathcal{R}_{\text{eq}}^{(1)}\left(\Sigma, \theta_0, \theta_{\text{pop}}^*\right) = \left\|(\Sigma + \tau I_p)^{-1}\Sigma(\theta_{\text{pop}}^* + D\theta_0) - \theta_{\text{pop}}^*\right\|_\Sigma^2$$

$$+ \kappa \operatorname{Tr}\left[\Sigma^2\left(\Sigma + \tau I_p\right)^{-2}\right]\frac{\sigma^2 + \tau^2\left\|(\Sigma + \tau I_p)^{-1}\left(\theta_{\text{pop}}^* + D\theta_0\right)\right\|_\Sigma^2}{p - \kappa \operatorname{Tr}\left[\Sigma^2\left(\Sigma + \tau I_p\right)^{-2}\right]}, \quad (30)$$

*and $\tau$ is the unique solution of (17).*

*Proof.* Note that we are generating labels using $\theta_0^{\text{perfo}} := \theta_{\text{pop}}^* + D\theta_0$ as a vector of regression coefficients. Thus, we can apply Theorem 3 by Ildiz et al. (2025) (which utilizes the non-asymptotic characterization of the minimum norm interpolator by Han & Xu (2023)), replacing $\beta^s$ with $\theta_0^{\text{perfo}}$ in that statement. This gives that (29) holds with $\mathcal{R}_{\text{eq}}^{(1)}\left(\Sigma, \theta_0, \theta_{\text{pop}}^*\right)$ replaced by $\widetilde{\mathcal{R}}_{\text{eq}}^{(1)}\left(\Sigma, \theta_{\text{pop}}^*, \theta_0^{\text{perfo}}\right)$ defined as

$$\widetilde{\mathcal{R}}_{\text{eq}}^{(1)}\left(\Sigma, \theta_{\text{pop}}^*, \theta_0^{\text{perfo}}\right) = \mathbb{E}_{g^{(1)}}\left[\left\|X^{(1)}\left(\Sigma, \theta_0^{\text{perfo}}, g^{(1)}\right) - \theta_{\text{pop}}^*\right\|_\Sigma^2\right], \quad (31)$$

where

$$X^{(1)}\left(\Sigma, \theta_0^{\text{perfo}}, g^{(1)}\right) = (\Sigma + \tau I_p)^{-1}\Sigma\left[\theta_0^{\text{perfo}} + \frac{\Sigma^{-1/2}\gamma^{(1)}(\theta_0^{\text{perfo}})g^{(1)}}{\sqrt{p}}\right], \quad (32)$$

$$\left(\gamma^{(1)}(\theta_0^{\text{perfo}})\right)^2 = \kappa\left(\sigma^2 + \widetilde{\mathcal{R}}_{\text{eq}}^{(1)}\left(\Sigma, \theta_0^{\text{perfo}}, \theta_0^{\text{perfo}}\right)\right), \quad (33)$$

$\tau$ is the unique solution of (17) and $g^{(1)} \sim \mathcal{N}(0, I_p)$. By plugging (32) into (31) and computing the expectation with respect to $g^{(1)}$, we get

$$\widetilde{\mathcal{R}}_{\text{eq}}^{(1)}\left(\Sigma, \theta_{\text{pop}}^*, \theta_0^{\text{perfo}}\right) = \left\|(\Sigma + \tau I_p)^{-1}\Sigma\theta_0^{\text{perfo}} - \theta_{\text{pop}}^*\right\|_\Sigma^2 + \frac{\left(\gamma^{(1)}(\theta_0^{\text{perfo}})\right)^2}{p}\operatorname{Tr}\left[\Sigma^2\left(\Sigma + \tau I_p\right)^{-2}\right]. \quad (34)$$

Next, we solve the fixed point equation in $\gamma^{(1)}(\theta_0^{\text{perfo}})$:

$$\left(\gamma^{(1)}(\theta_0^{\text{perfo}})\right)^2 = \kappa\left(\sigma^2 + \widetilde{\mathcal{R}}_{\text{eq}}^{(1)}\left(\Sigma, \theta_0^{\text{perfo}}, \theta_0^{\text{perfo}}\right)\right)$$

$$= \kappa\left(\sigma^2 + \left\|\left((\Sigma + \tau I_p)^{-1}\Sigma - I_p\right)\theta_0^{\text{perfo}}\right\|_\Sigma^2 + \frac{\left(\gamma^{(1)}(\theta_0^{\text{perfo}})\right)^2}{p}\operatorname{Tr}\left[\Sigma^2\left(\Sigma + \tau I_p\right)^{-2}\right]\right)$$

$$= \kappa\left(\sigma^2 + \tau^2\left\|(\Sigma + \tau I_p)^{-1}\theta_0^{\text{perfo}}\right\|_\Sigma^2 + \frac{\left(\gamma^{(1)}(\theta_0^{\text{perfo}})\right)^2}{p}\operatorname{Tr}\left[\Sigma^2\left(\Sigma + \tau I_p\right)^{-2}\right]\right).$$

The last equality comes from

$$I_p - (\Sigma + \tau I_p)^{-1}\Sigma = (\Sigma + \tau I_p)^{-1}\left(\Sigma + \tau I_p\right) - (\Sigma + \tau I_p)^{-1}\Sigma = \tau\left(\Sigma + \tau I_p\right)^{-1}.$$

Rearranging gives that

$$\left(\gamma^{(1)}(\theta_0^{\text{perfo}})\right)^2 = \kappa\frac{\sigma^2 + \tau^2\left\|(\Sigma + \tau I_p)^{-1}\theta_0^{\text{perfo}}\right\|_\Sigma^2}{1 - \frac{1}{n}\operatorname{Tr}\left[\Sigma^2\left(\Sigma + \tau I_p\right)^{-2}\right]}, \quad (35)$$

which plugged into (34) gives the desired result. $\qquad\square$

We note that the expression in (30) depends on $\theta_0$ and, in fact, it keeps depending on it even after neglecting terms of order $O(\|D\|_{\text{op}}^2)$.

**Deterministic equivalent for** $\mathcal{R}_2(\Sigma, \theta_2, \theta_{\text{pop}}^*)$**.** Next, by iterating twice the strategy of Lemma 7, we derive a deterministic equivalent for $\mathcal{R}_2(\Sigma, \theta_2, \theta_{\text{pop}}^*)$.

**Lemma 8.** *Let Assumption 1 hold. Let $R > 0$ be a constant such that $\theta_{\text{pop}}^*, \theta_0 \in B_p(R)$. Assume that $\kappa, \sigma, \lambda \in (1/M, M)$ and $\|\Sigma\|_{\text{op}}, \|\Sigma^{-1}\|_{\text{op}} \leq M$ for some constant $M > 1$. Then, there exists a constant $C = C(M, R)$ such that, for any $\delta \in (0, 1/2]$, the following holds*

$$\sup_{\theta_{\text{pop}}^*, \theta_0 \in B_p(R)} \Pr\left(\left|\mathcal{R}_2(\Sigma, \theta_2, \theta_{\text{pop}}^*) - \mathcal{R}_{\text{eq}}^{(2)}\left(\Sigma, \theta_0, \theta_{\text{pop}}^*\right)\right| \geq \delta\right) \leq Cpe^{-p\delta^4/C}, \qquad (36)$$

*with probability at least $1 - Cpe^{-p\delta^4/C}$, where*

$$\mathcal{R}_{\text{eq}}^{(2)}\left(\Sigma, \theta_0, \theta_{\text{pop}}^*\right) = \left\|(\Sigma + \tau I_p)^{-1}\Sigma D(\Sigma + \tau I_p)^{-1}\Sigma(\theta_{\text{pop}}^* + D\theta_0) - \tau(\Sigma + \tau I_p)^{-1}\theta_{\text{pop}}^*\right\|_\Sigma^2$$

$$+ \kappa \operatorname{Tr}\left[\Sigma(\Sigma + \tau I_p)^{-2}D\Sigma^3(\Sigma + \tau I_p)^{-2}D\right]\frac{\sigma^2 + \tau^2\left\|(\Sigma + \tau I_p)^{-1}(\theta_{\text{pop}}^* + D\theta_0)\right\|_\Sigma^2}{p - \kappa\operatorname{Tr}\left[\Sigma^2(\Sigma + \tau I_p)^{-2}\right]}$$

$$+ \kappa\operatorname{Tr}\left[\Sigma^2(\Sigma + \tau I_p)^{-2}\right]\frac{\sigma^2 + \tau^2\left\|(\Sigma + \tau I_p)^{-1}\left(\theta_{\text{pop}}^* + D(\Sigma + \tau I_p)^{-1}\Sigma(\theta_{\text{pop}}^* + D\theta_0)\right)\right\|_\Sigma^2}{p - \kappa\operatorname{Tr}\left[\Sigma^2(\Sigma + \tau I_p)^{-2}\right]}$$

$$+ \kappa^2\tau^2\operatorname{Tr}\left[\Sigma^2(\Sigma + \tau I_p)^{-2}\right]\operatorname{Tr}\left[\Sigma(\Sigma + \tau I_p)^{-2}D\Sigma(\Sigma + \tau I_p)^{-2}D\right]$$

$$\cdot\frac{\sigma^2 + \tau^2\left\|(\Sigma + \tau I_p)^{-1}(\theta_{\text{pop}}^* + D\theta_0)\right\|_\Sigma^2}{\left(p - \kappa\operatorname{Tr}\left[\Sigma^2(\Sigma + \tau I_p)^{-2}\right]\right)^2},$$

$$(37)$$

*and $\tau$ is the unique solution of* (17).

*Proof.* The proof extends the argument of Theorem 2 by (Ildiz et al., 2025) to the ridge regression case, and it applies the distributional characterization of the minimum norm interpolator by Han & Xu (2023) twice. First, note that $\|\theta_1\|_2$ is bounded by a constant $C_1 = C_1(R, M)$ independent of $n, p$, with probability at least $C_2e^{-p/C_2}$, where $C_2 = C_2(R, M)$ is a constant independent of $n, p$. This follows from a direct adaptation of Proposition 11 by Ildiz et al. (2025). Define $R' := \max(C_1, \|\theta_{\text{pop}}^*\|_2)$. Then, upon conditioning on $\theta_1$, we can apply Lemma 7 (after re-defining $R$ to be $R'$), which gives that, for some constant $C_3 = C_3(R, M)$,

$$\sup_{\theta_{\text{pop}}^*, \theta_1 \in B_p(R')} \Pr\left(\left|\mathcal{R}_2(\Sigma, \theta_2, \theta_{\text{pop}}^*) - \mathcal{R}_{\text{eq}}^{(1)}\left(\Sigma, \theta_1, \theta_{\text{pop}}^*\right)\right| \geq \delta\right) \leq C_3pe^{-p\delta^4/C_3}, \qquad (38)$$

where $\mathcal{R}_{\text{eq}}^{(1)}\left(\Sigma, \theta_1, \theta_{\text{pop}}^*\right)$ is defined in (30) and $\tau$ is the unique solution of (17).

We now evaluate the first term in the expression for $\mathcal{R}_{\text{eq}}^{(1)}\left(\Sigma, \theta_1, \theta_{\text{pop}}^*\right)$:

$$\mathcal{R}_{\text{eq},1}^{(1)}\left(\Sigma, \theta_1, \theta_{\text{pop}}^*\right) := \left\|(\Sigma + \tau I_p)^{-1}\Sigma(\theta_{\text{pop}}^* + D\theta_1) - \theta_{\text{pop}}^*\right\|_\Sigma^2$$

$$= \left\|(\Sigma + \tau I_p)^{-1}\Sigma D\theta_1 - \tau(\Sigma + \tau I_p)^{-1}\theta_{\text{pop}}^*\right\|_\Sigma^2.$$

Let $M_1 = \Sigma^{1/2}$, $M_2 = (\Sigma + \tau I_p)^{-1}\Sigma D$ and $a = \tau(\Sigma + \tau I_p)^{-1}\theta_{\text{pop}}^*$. Then, the function $\theta_1 \mapsto \mathcal{R}_{\text{eq},1}^{(1)}\left(\Sigma, \theta_1, \theta_{\text{pop}}^*\right)$ can be expressed as

$$f(\theta_1) = \|M_1(M_2\theta_1 - a)\|_2^2,$$

which has gradient

$$\nabla f(\theta_1) = 2M_2^\top M_1^\top M_1(M_2\theta_1 - a).$$

As $\|\theta_1\|_2 \leq C_1$, $f$ is Lipschitz and its Lipschitz constant is $2\|M_1\|_{\text{op}}^2\|M_2\|_{\text{op}}(\|M_1\|_{\text{op}}C_1 + \|a\|_2)$. As $\|M_1\|_{\text{op}}, \|M_2\|_{\text{op}}, C_1, \|a\|_2$ are all upper bounded by constants dependent only on $R, M$, the Lipschitz constant of $f$ is also upper bounded by a constant dependent only on $R, M$. Thus, an

application of the distributional characterization by Han & Xu (2023) (restated as Theorem 4 in (Ildiz et al., 2025)) gives that, for some constant $C_4 = C_4(R, M)$,

$$\sup_{\theta^*_{\text{pop}}, \theta_1 \in B_p(R')} \Pr\left(\left|\mathcal{R}^{(1)}_{\text{eq},1}\left(\Sigma, \theta_1, \theta^*_{\text{pop}}\right) - \widetilde{\mathcal{R}}^{(1)}_{\text{eq},1}\left(\Sigma, \theta^*_{\text{pop}}, \theta^{\text{perfo}}_0\right)\right| \geq \delta\right) \leq C_4 p e^{-p\delta^4/C_4}, \quad (39)$$

where

$$\widetilde{\mathcal{R}}^{(1)}_{\text{eq},1}\left(\Sigma, \theta^*_{\text{pop}}, \theta^{\text{perfo}}_0\right) = \mathbb{E}_{g^{(1)}}\left[\left\|(\Sigma + \tau I_p)^{-1}\Sigma D X^{(1)}\left(\Sigma, \theta^{\text{perfo}}_0, g^{(1)}\right) - \tau(\Sigma + \tau I_p)^{-1}\theta^*_{\text{pop}}\right\|^2_\Sigma\right]. \quad (40)$$

We recall from Lemma 7 that $\theta^{\text{perfo}}_0 = \theta^*_{\text{pop}} + D\theta_0$, $g^{(1)} \sim \mathcal{N}(0, I_p)$ and $X^{(1)}\left(\Sigma, \theta^{\text{perfo}}_0, g^{(1)}\right)$ is given by (32). By plugging (32) into the RHS of (40) and computing the expectation with respect to $g^{(1)}$, we have

$$\widetilde{\mathcal{R}}^{(1)}_{\text{eq},1}\left(\Sigma, \theta^*_{\text{pop}}, \theta^{\text{perfo}}_0\right) = \mathbb{E}_{g^{(1)}}\left[\left\|(\Sigma + \tau I_p)^{-1}\Sigma D(\Sigma + \tau I_p)^{-1}\Sigma\theta^{\text{perfo}}_0 - \tau(\Sigma + \tau I_p)^{-1}\theta^*_{\text{pop}}\right.\right.$$

$$\left.\left. + (\Sigma + \tau I_p)^{-1}\Sigma D(\Sigma + \tau I_p)^{-1}\Sigma^{1/2}\frac{\gamma^{(1)}(\theta^{\text{perfo}}_0)g^{(1)}}{\sqrt{p}}\right\|^2_\Sigma\right]$$

$$= \left\|(\Sigma + \tau I_p)^{-1}\Sigma D(\Sigma + \tau I_p)^{-1}\Sigma\theta^{\text{perfo}}_0 - \tau(\Sigma + \tau I_p)^{-1}\theta^*_{\text{pop}}\right\|^2_\Sigma$$

$$+ \frac{\left(\gamma^{(1)}(\theta^{\text{perfo}}_0)\right)^2}{p}\text{Tr}\left[\Sigma\left(\Sigma + \tau I_p\right)^{-2}D\Sigma^3\left(\Sigma + \tau I_p\right)^{-2}D\right], \quad (41)$$

where in the last step we have used the circulant property of the trace. By using the expression for $\gamma^{(1)}(\theta^{\text{perfo}}_0)$ in (35) and recalling that $\theta^{\text{perfo}}_0 = \theta^*_{\text{pop}} + D\theta_0$, one readily obtains that the RHS of (41) coincides with the first two lines of the RHS of (37).

Finally, we evaluate the second term in the expression for $\mathcal{R}^{(1)}_{\text{eq}}\left(\Sigma, \theta_1, \theta^*_{\text{pop}}\right)$:

$$\mathcal{R}^{(1)}_{\text{eq},2}\left(\Sigma, \theta_1, \theta^*_{\text{pop}}\right) := \kappa\,\text{Tr}\left[\Sigma^2\left(\Sigma + \tau I_p\right)^{-2}\right]\frac{\sigma^2 + \tau^2\left\|(\Sigma + \tau I_p)^{-1}\left(\theta^*_{\text{pop}} + D\theta_1\right)\right\|^2_\Sigma}{p - \kappa\,\text{Tr}\left[\Sigma^2\left(\Sigma + \tau I_p\right)^{-2}\right]}.$$

Let $M_1 = \Sigma^{1/2}$, $M_2 = (\Sigma + \tau I_p)^{-1}D$ and $a = (\Sigma + \tau I_p)^{-1}\theta^*_{\text{pop}}$. Then, the function $\theta_1 \mapsto \left\|(\Sigma + \tau I_p)^{-1}\left(\theta^*_{\text{pop}} + D\theta_1\right)\right\|^2_\Sigma$ can be expressed as

$$f(\theta_1) = \|M_1(M_2\theta_1 + a)\|^2_2,$$

which has gradient

$$\nabla f(\theta_1) = 2M_2^\top M_1^\top M_1(M_2\theta_1 + a).$$

As $\|\theta_1\|_2 \leq C_1$, $f$ is Lipschitz and its Lipschitz constant is $2\|M_1\|^2_{\text{op}}\|M_2\|_{\text{op}}(\|M_1\|_{\text{op}}C_1 + \|a\|_2)$. As $\|M_1\|_{\text{op}}, \|M_2\|_{\text{op}}, C_1, \|a\|_2$ are all upper bounded by constants dependent only on $R, M$, the Lipschitz constant of $f$ is also upper bounded by a constant dependent only on $R, M$. Note that the quantity $\left|p - \kappa\,\text{Tr}\left[\Sigma^2(\Sigma + \tau I_p)^{-2}\right]\right|$ is lower bounded by a constant dependent only on $R, M$, as a consequence of Proposition 2.1 in Han & Xu (2023). Thus, we have that the function $\theta_1 \mapsto \mathcal{R}^{(1)}_{\text{eq},2}\left(\Sigma, \theta_1, \theta^*_{\text{pop}}\right)$ is Lipschitz and its Lipschitz constant is $C_5 = C_5(R, M)$. Hence, another application of the distributional characterization by Han & Xu (2023) (cf. Theorem 4 in (Ildiz et al., 2025)) gives that, for some constant $C_6 = C_6(R, M)$,

$$\sup_{\theta^*_{\text{pop}}, \theta_1 \in B_p(R')} \Pr\left(\left|\mathcal{R}^{(1)}_{\text{eq},2}\left(\Sigma, \theta_1, \theta^*_{\text{pop}}\right) - \widetilde{\mathcal{R}}^{(1)}_{\text{eq},2}\left(\Sigma, \theta^*_{\text{pop}}, \theta^{\text{perfo}}_0\right)\right| \geq \delta\right) \leq C_6 p e^{-p\delta^4/C_6}, \quad (42)$$

where

$$\widetilde{\mathcal{R}}_{\text{eq},2}^{(1)} \left( \Sigma, \theta_{\text{pop}}^*, \theta_0^{\text{perfo}} \right) = \kappa \operatorname{Tr} \left[ \Sigma^2 \left( \Sigma + \tau I_p \right)^{-2} \right]$$

$$\cdot \frac{\sigma^2 + \tau^2 \mathbb{E}_{g^{(1)}} \left[ \left\| \left( \Sigma + \tau I_p \right)^{-1} \left( \theta_{\text{pop}}^* + DX^{(1)} \left( \Sigma, \theta_0^{\text{perfo}}, g^{(1)} \right) \right) \right\|_\Sigma^2 \right]}{p - \kappa \operatorname{Tr} \left[ \Sigma^2 \left( \Sigma + \tau I_p \right)^{-2} \right]}. \tag{43}$$

By using (32) and computing the expectation with respect to $g^{(1)}$, we have

$$\mathbb{E}_{g^{(1)}} \left[ \left\| \left( \Sigma + \tau I_p \right)^{-1} \left( \theta_{\text{pop}}^* + DX^{(1)} \left( \Sigma, \theta_0^{\text{perfo}}, g^{(1)} \right) \right) \right\|_\Sigma^2 \right]$$

$$= \mathbb{E}_{g^{(1)}} \left[ \left\| \left( \Sigma + \tau I_p \right)^{-1} \theta_{\text{pop}}^* + \left( \Sigma + \tau I_p \right)^{-1} D \left( \Sigma + \tau I_p \right)^{-1} \Sigma \theta_0^{\text{perfo}} \right. \right.$$

$$\left. \left. + \left( \Sigma + \tau I_p \right)^{-1} D \left( \Sigma + \tau I_p \right)^{-1} \Sigma^{1/2} \frac{\gamma^{(1)}(\theta_0^{\text{perfo}}) g^{(1)}}{\sqrt{p}} \right\|_\Sigma^2 \right] \tag{44}$$

$$= \left\| \left( \Sigma + \tau I_p \right)^{-1} \theta_{\text{pop}}^* + \left( \Sigma + \tau I_p \right)^{-1} D \left( \Sigma + \tau I_p \right)^{-1} \Sigma \theta_0^{\text{perfo}} \right\|_\Sigma^2$$

$$+ \frac{\left( \gamma^{(1)}(\theta_0^{\text{perfo}}) \right)^2}{p} \operatorname{Tr} \left[ \Sigma \left( \Sigma + \tau I_p \right)^{-2} D \Sigma \left( \Sigma + \tau I_p \right)^{-2} D \right],$$

where in the last step we have used the circulant property of the trace. By plugging (44) into (43), using the expression for $\gamma^{(1)}(\theta_0^{\text{perfo}})$ in (35) and recalling that $\theta_0^{\text{perfo}} = \theta_{\text{pop}}^* + D\theta_0$, one readily obtains that the RHS of (43) coincides with the last two lines of the RHS of (37). As $\mathcal{R}_{\text{eq}}^{(1)} \left( \Sigma, \theta_1, \theta_{\text{pop}}^* \right) = \mathcal{R}_{\text{eq},1}^{(1)} \left( \Sigma, \theta_1, \theta_{\text{pop}}^* \right) + \mathcal{R}_{\text{eq},2}^{(1)} \left( \Sigma, \theta_1, \theta_{\text{pop}}^* \right)$, the desired result readily follows by combining (38), (39) and (42). □

**Concluding the argument.** Note that

$$\operatorname{Tr} \left[ \Sigma (\Sigma + \tau I_p)^{-2} D \Sigma^3 (\Sigma + \tau I_p)^{-2} D \right] = O(\|D\|_{\text{op}}^2),$$

$$\operatorname{Tr} \left[ \Sigma (\Sigma + \tau I_p)^{-2} D \Sigma (\Sigma + \tau I_p)^{-2} D \right] = O(\|D\|_{\text{op}}^2).$$

Furthermore, we have

$$\left\| \left( \Sigma + \tau I_p \right)^{-1} \Sigma D \left( \Sigma + \tau I_p \right)^{-1} \Sigma (\theta_{\text{pop}}^* + D\theta_0) - \tau (\Sigma + \tau I_p)^{-1} \theta_{\text{pop}}^* \right\|_\Sigma^2$$

$$= \left\| \left( \Sigma + \tau I_p \right)^{-1} \Sigma D \left( \Sigma + \tau I_p \right)^{-1} \Sigma \theta_{\text{pop}}^* - \tau (\Sigma + \tau I_p)^{-1} \theta_{\text{pop}}^* \right\|_\Sigma^2 + O(\|D\|_{\text{op}}^2)$$

$$= \left\| \tau (\Sigma + \tau I_p)^{-1} \theta_{\text{pop}}^* \right\|_\Sigma^2$$

$$\quad - 2\tau \langle (\Sigma + \tau I_p)^{-1} \theta_{\text{pop}}^*, \Sigma \left( \Sigma + \tau I_p \right)^{-1} \Sigma D \left( \Sigma + \tau I_p \right)^{-1} \Sigma \theta_{\text{pop}}^* \rangle + O(\|D\|_{\text{op}}^2)$$

$$= \tau \langle \theta_{\text{pop}}^*, (\Sigma + \tau I_p)^{-1} \left( \tau I_p - 2(\Sigma + \tau I_p)^{-1} \Sigma^2 D \right) \Sigma (\Sigma + \tau I_p)^{-1} \theta_{\text{pop}}^* \rangle + O(\|D\|_{\text{op}}^2).$$

Similarly, we have

$$\left\| \left( \Sigma + \tau I_p \right)^{-1} \left( \theta_{\text{pop}}^* + D \left( \Sigma + \tau I_p \right)^{-1} \Sigma (\theta_{\text{pop}}^* + D\theta_0) \right) \right\|_\Sigma^2$$

$$= \left\| \left( \Sigma + \tau I_p \right)^{-1} \left( \theta_{\text{pop}}^* + D \left( \Sigma + \tau I_p \right)^{-1} \Sigma \theta_{\text{pop}}^* \right) \right\|_\Sigma^2 + O(\|D\|_{\text{op}}^2)$$

$$= \left\| \left( \Sigma + \tau I_p \right)^{-1} \theta_{\text{pop}}^* \right\|_\Sigma^2 + 2 \langle (\Sigma + \tau I_p)^{-1} \theta_{\text{pop}}^*, \Sigma \left( \Sigma + \tau I_p \right)^{-1} D \left( \Sigma + \tau I_p \right)^{-1} \Sigma \theta_{\text{pop}}^* \rangle + O(\|D\|_{\text{op}}^2)$$

$$= \langle \theta_{\text{pop}}^*, (\Sigma + \tau I_p)^{-1} \left( I_p + 2 \left( \Sigma + \tau I_p \right)^{-1} \Sigma D \right) \Sigma \left( \Sigma + \tau I_p \right)^{-1} \theta_{\text{pop}}^* \rangle + O(\|D\|_{\text{op}}^2).$$

Recalling the definitions (16) and (37), we conclude that

$$\mathcal{R}_{\text{eq}}^{(2)} \left( \Sigma, \theta_0, \theta_{\text{pop}}^* \right) = \mathcal{R}_{\text{eq}} \left( \Sigma, \theta_{\text{pop}}^*, D, \lambda \right) + O(\|D\|_{\text{op}}^2). \tag{45}$$

Thus, the desired result follows from (45) and Lemma 8.

## B.1 EXTENSION TO SUB-GAUSSIAN DATA

Throughout this appendix, we relax Assumption 1 as follows.

**Assumption 2** (Regression performative model – relaxed assumption). For $\theta \in \mathbb{R}^p$, samples from $\mathcal{D}(\theta)$ are taken i.i.d. with features $x$ drawn independently of $\theta$ and such that $\Sigma^{-1/2}x$ has independent, zero mean, unit variance and uniformly sub-Gaussian entries. The label $y$ is given by

$$y = x^\top \theta_{\text{pop}}^* + x^\top D\theta + w, \quad w \sim \mathcal{N}(0, \sigma^2). \tag{46}$$

We assume $p = 2d$, $(\theta_{\text{pop}}^*)^\top = (a^\top, 0)$ with $a$ having zero mean and covariance $I_d/d$, and $D = \text{diag}(b, c)$ where $b, c \in \mathbb{R}^d$ with $\|b\|_\infty, \|c\|_\infty < 1$. We further assume that $a\sqrt{d}$ has sub-Gaussian norm upper bounded by a universal constant (independent of $d$).

**Theorem 9** (Excess risk – over-parameterized, relaxed assumptions). *Let Assumption 2 hold. Let $R > 0$ be a constant s.t. $\theta_{\text{pop}}^* \in B_p(R)$ and let $\theta_0$ be sampled uniformly on the unit sphere. Assume that $\kappa, \sigma, \lambda \in (1/M, M)$ and $\|\Sigma\|_{\text{op}}, \|\Sigma^{-1}\|_{\text{op}} \leq M$ for some constant $M > 1$. Then, there exists a constant $C = C(M, R)$ such that for any $\delta \in (0, 1/2]$, with probability at least $1 - C\delta^{-7}p^{-1/8}$,*

$$\left|\mathcal{R}(\Sigma, \theta_2, \theta_{\text{pop}}^*) - \mathcal{R}_{\text{eq}}\left(\Sigma, \theta_{\text{pop}}^*, D, \lambda\right)\right| \leq \delta + O(\|D\|_{\text{op}}^2), \tag{47}$$

*where $\mathcal{R}_{\text{eq}}\left(\Sigma, \theta_{\text{pop}}^*, D, \lambda\right)$ is given by (16).*

**Lemma 10** (Norm control). *In the setting of Theorem 9, we have that*

$$\|\theta_1\|_2 \leq C, \tag{48}$$

$$\|\theta_2\|_2 \leq C, \tag{49}$$

$$\|\theta_0\|_\infty \leq C\frac{\log p}{\sqrt{p}}, \tag{50}$$

$$\|\theta_{\text{pop}}^*\|_\infty \leq C\frac{\log p}{\sqrt{p}}, \tag{51}$$

$$\|\theta_1\|_\infty \leq C\frac{\log p}{\sqrt{p}}, \tag{52}$$

*with probability at least $1 - Ce^{-\log^2 p/C}$, where $C = C(R, M)$ is a constant depending only on $R, M$ (and not on $n, p$).*

*Proof.* We start by proving (48). The claim follows by extending the argument of Proposition 11 in Ildiz et al. (2025) and we repeat it here for completeness. Recall that

$$\theta_1 = \frac{1}{p}\left(\frac{1}{p}X^{0\top}X^0 + \lambda I_p\right)^{-1}X^{0\top}X^0(\theta_{\text{pop}}^* + D\theta_0) + \frac{1}{p}\left(\frac{1}{p}X^{0\top}X^0 + \lambda I_p\right)^{-1}X^{0\top}w.$$

Note that

$$\left\|\frac{1}{p}\left(\frac{1}{p}X^{0\top}X^0 + \lambda I_p\right)^{-1}X^{0\top}X^0\right\|_{\text{op}} \leq 1,$$

which implies that

$$\left\|\frac{1}{p}\left(\frac{1}{p}X^{0\top}X^0 + \lambda I_p\right)^{-1}X^{0\top}X^0(\theta_{\text{pop}}^* + D\theta_0)\right\|_2 \leq C_1,$$

for some constant $C_1 = C_1(R, M)$. Next, we can write

$$\left\|\frac{1}{p}\left(\frac{1}{p}X^{0\top}X^0 + \lambda I_p\right)^{-1}X^{0\top}w\right\|_2^2 = \frac{w^\top X^0}{p}\left(\frac{1}{p}X^{0\top}X^0 + \lambda I_p\right)^{-2}\frac{X^{0\top}w}{p}$$

$$\leq \frac{w^\top w}{p}\left\|\frac{1}{p}X^0\left(\frac{1}{p}X^{0\top}X^0 + \lambda I_p\right)^{-2}X^{0\top}\right\|_{\text{op}}.$$

Using Bernstein's inequality, we have that $w^\top w/p$ is upper bounded by $C_2 = C_2(R,M)$ with probability at least $1 - e^{-p/C_2}$. Furthermore, $\left\|\frac{1}{p}X^0\left(\frac{1}{p}X^{0\top}X^0 + \lambda I_p\right)^{-2}X^{0\top}\right\|_{\text{op}} \leq$

$\left\|\frac{1}{p}X^0\left(\frac{1}{p}X^{0\top}X^0 + \lambda I_p\right)^{-2}X^{0\top}\right\|_{\text{op}}$ is also upper bounded by a universal constant. Thus, an application of the triangle inequality gives (48). Repeating the same argument with $\theta_1$ in place of $\theta_0$ and $X^1$ in place of $X^0$ readily gives (49).

Let $v \in \mathbb{R}^p$ be a vector such that $v\sqrt{p}$ has sub-Gaussian norm upper bounded by a universal constant (independent of $p$). We will now show that

$$\|v\|_\infty \leq C\frac{\log p}{\sqrt{p}}, \tag{53}$$

with probability at least $1 - e^{-\log^2 p}$. To see this, it suffices to note that the $j$-th coordinate $v_j\sqrt{p}$ is sub-Gaussian with sub-Gaussian norm upper bounded by a universal constant. Thus,

$$\mathbb{P}(|v_j\sqrt{p}| > t) \leq 2e^{-t^2/C_3},$$

for some universal constant $C_3$. Taking $t = C\log p$ and doing a union bound over $j \in \{1,\ldots,p\}$ gives (53).

Since $\theta_0$ is sampled uniformly on the sphere, (50) is implied by (53). Therefore, $\theta_0\sqrt{p}$ has sub-Gaussian norm upper bounded by a universal constant (independent of $p$). Furthermore, (51) is implied by (53) since $\theta_{\text{pop}}^*$ satisfies Assumption 2. Finally, letting $\|\cdot\|_{\psi_2}$ denote the sub-Gaussian norm of a vector, we have

$$\|\theta_1\sqrt{p}\|_{\psi_2} \leq \left\|\frac{1}{p}\left(\frac{1}{p}X^{0\top}X^0 + \lambda I_p\right)^{-1}X^{0\top}X^0\right\|_{\text{op}}(\|\theta_{\text{pop}}^*\sqrt{p}\|_{\psi_2} + \|\theta_0\sqrt{p}\|_{\psi_2})$$

$$+ \left\|\frac{1}{\sqrt{p}}\left(\frac{1}{p}X^{0\top}X^0 + \lambda I_p\right)^{-1}X^{0\top}\right\|_{\text{op}}\|w\|_{\psi_2} \tag{54}$$

$$\leq \|\theta_{\text{pop}}^*\sqrt{p}\|_{\psi_2} + \|\theta_0\sqrt{p}\|_{\psi_2} + \|w\|_{\psi_2},$$

which is upper bounded by a universal constant. Thus, (52) is also implied by (53) and the proof is complete. □

**Lemma 11.** *Let Assumption 2 hold. Let $R > 0$ be a constant such that $\theta_{\text{pop}}^* \in B_p(R)$ and let $\theta_0$ be sampled uniformly on the unit sphere. Assume that $\kappa, \sigma, \lambda \in (1/M, M)$ and $\|\Sigma\|_{\text{op}}, \|\Sigma^{-1}\|_{\text{op}} \leq M$ for some constant $M > 1$. Then, there exists a constant $C = C(M,R)$ such that, for any $\delta \in (0, 1/2]$, the following holds*

$$\sup_{\theta_{\text{pop}}^*, \theta_0 \in B_p(R)} \Pr\left(\left|\mathcal{R}_1(\Sigma, \theta_1, \theta_{\text{pop}}^*) - \mathcal{R}_{\text{eq}}^{(1)}\left(\Sigma, \theta_0, \theta_{\text{pop}}^*\right)\right| \geq \delta\right) \leq Cpe^{-p\delta^4/C}, \tag{55}$$

*with probability at least $1 - C\delta^{-7}p^{-1/8}$, where $\mathcal{R}_{\text{eq}}^{(1)}$ is given by (30).*

*Proof.* By Lemma 10, we have that $\theta_{\text{pop}}^* + D\theta_0$ satisfies the delocalization condition of Proposition 10.3 by Han & Xu (2023). This implies that the hypotheses of Theorem 2.4 by Han & Xu (2023) are satisfied when we train using $\theta_{\text{pop}}^* + D\theta_0$. Thus, we can now follow the same steps as in Lemma 7 which invokes Theorem 3 by Ildiz et al. (2025). In particular, Theorem 3 by Ildiz et al. (2025) uses Theorem 4 therein plus the bound on $\|\theta_1\|_2$ given by Lemma 10. Thus, it suffices to replace the application of Theorem 4 by Ildiz et al. (2025) with the application of Theorem 2.4 by Han & Xu (2023), and the desired result readily holds. □

*Proof of Theorem 9.* By Lemma 10, we have that $\theta_{\text{pop}}^* + D\theta_0$ and $\theta_{\text{pop}}^* + D\theta_1$ satisfy the delocalization condition of Proposition 10.3 by Han & Xu (2023). This implies that the hypotheses of Theorem 2.4 by Han & Xu (2023) are satisfied when we train using either $\theta_{\text{pop}}^* + D\theta_0$ or $\theta_{\text{pop}}^* + D\theta_1$ as vector of regression coefficients. Consequently, the desired result is obtained by following the same steps as in the proof of Theorem 3, the only differences being that *(i)* we apply Lemma 11 in place of Lemma 7, and *(ii)* we apply Theorem 2.4 by Han & Xu (2023) in place of Theorem 4 by Ildiz et al. (2025). This requires an upper bound on $\|\theta_1\|_2, \|\theta_2\|_2$ which is provided by Lemma 10. □

## C  PROOF OF THEOREM 4

We start by computing explicitly $\mathbb{E}_{\theta_{\text{pop}}^*} \mathcal{R}_{\text{eq}}\left(\Sigma, \theta_{\text{pop}}^*, D, \lambda\right)$.

**Lemma 12.** *Consider the setting of Theorem 3, assume that $a$ has covariance $I_d/d$, and let $\Sigma = \begin{bmatrix} I_d & \rho I_d \\ \rho I_d & I_d \end{bmatrix}$. Then, we have that*

$$\mathbb{E}_{\theta_{\text{pop}}^*} \mathcal{R}_{\text{eq}}\left(\Sigma, \theta_{\text{pop}}^*, D, \lambda\right) = \widetilde{\mathcal{R}}(D, \lambda, \rho) + O(\bar{b}\rho^2 + \rho^4),$$

$$\widetilde{\mathcal{R}}(D, \lambda, \rho) := \mathcal{R}_0(\lambda, \rho) + \bar{b} A_1(\lambda) + \bar{c}\rho^2 A_2(\lambda),$$

*where $\bar{b} = \text{Tr}[\text{diag}(b)]/d, \bar{c} = \text{Tr}[\text{diag}(c)]/d$ and the auxiliary functions $\mathcal{R}_0(\lambda, \rho)$, $A_1(\lambda)$, and $A_2(\lambda)$ are given by*

$$\mathcal{R}_0(\lambda, \rho) = \frac{\tau^2}{(1+\tau)^2} + \frac{\kappa}{(1+\tau)^2 - \kappa}\left(\sigma^2 + \frac{\tau^2}{(1+\tau)^2}\right)$$

$$+ \rho^2 \left(\frac{\tau^2(1-2\tau)}{(1+\tau)^4} + \frac{\kappa\tau^2(1-2\tau)}{(1+\tau)^4\left((1+\tau)^2 - \kappa\right)} + \frac{\kappa\tau(\tau-2)}{\left((1+\tau)^2 - \kappa\right)^2}\left(\sigma^2 + \frac{\tau^2}{(1+\tau)^2}\right)\right),$$

$$A_1(\lambda) = -\frac{2\tau}{(1+\tau)^3} + \frac{2\kappa\tau^2}{(1+\tau)^3\left((1+\tau)^2 - \kappa\right)},$$

$$A_2(\lambda) = -\frac{4\tau^3}{(1+\tau)^5} + \frac{2\kappa\tau^3(\tau^2 - 1)}{(1+\tau)^6\left((1+\tau)^2 - \kappa\right)}.$$

$$(56)$$

*Proof.* Given a $p \times p$ matrix $M$, let us denote by $(M)_1$ its top-left $d \times d$ block. For any $M \in \mathbb{R}^{p \times p}$, we have

$$\mathbb{E}_{\theta_{\text{pop}}^*}\left[\langle\theta_{\text{pop}}^*, M\theta_{\text{pop}}^*\rangle\right] = \mathbb{E}_{\theta_{\text{pop}}^*}\left[(\theta_{\text{pop}}^*)^\top M\theta_{\text{pop}}^*\right] = \mathbb{E}_{\theta_{\text{pop}}^*}\left[\text{Tr}\left[(\theta_{\text{pop}}^*)^\top M\theta_{\text{pop}}^*\right]\right]$$

$$= \mathbb{E}_{\theta_{\text{pop}}^*}\left[\text{Tr}\left[M\theta_{\text{pop}}^*(\theta_{\text{pop}}^*)^\top\right]\right] = \text{Tr}\left[(M)_1\right]/d,$$

where the third equality uses the circulant property of the trace and the last one that $(\theta_{\text{pop}}^*)^\top = (a^\top, 0)$ with $a$ having covariance $I_d/d$. Thus, from (16), we have

$$\mathbb{E}_{\theta_{\text{pop}}^*} \mathcal{R}_{\text{eq}}\left(\Sigma, \theta_{\text{pop}}^*, D, \lambda\right) = \frac{\tau^2}{d} \text{Tr}\left[\left(\Sigma\left(\Sigma + \tau I_p\right)^{-2}\right)_1\right]$$

$$+ \kappa \text{Tr}\left[\Sigma^2\left(\Sigma + \tau I_p\right)^{-2}\right] \frac{\sigma^2 + \frac{\tau^2}{d}\text{Tr}\left[\left(\Sigma\left(\Sigma + \tau I_p\right)^{-2}\right)_1\right]}{p - \kappa\text{Tr}\left[\Sigma^2\left(\Sigma + \tau I_p\right)^{-2}\right]}$$

$$- \frac{2\tau}{d}\text{Tr}\left[\left(\left(\Sigma + \tau I_p\right)^{-2}\Sigma^2 D\left(\Sigma + \tau I_p\right)^{-1}\Sigma\right)_1\right]$$

$$+ \frac{2\kappa\tau^2}{d}\text{Tr}\left[\Sigma^2\left(\Sigma + \tau I_p\right)^{-2}\right] \frac{\text{Tr}\left[\left(\left(\Sigma + \tau I_p\right)^{-2}\Sigma D\left(\Sigma + \tau I_p\right)^{-1}\Sigma\right)_1\right]}{p - \kappa\text{Tr}\left[\Sigma^2\left(\Sigma + \tau I_p\right)^{-2}\right]}.$$

$$(57)$$

Note that $\Sigma + \tau I_p = \begin{bmatrix} 1+\tau & \rho \\ \rho & 1+\tau \end{bmatrix} \otimes I_d$ has inverse $(\Sigma + \tau I_p)^{-1} = \frac{1}{(1+\tau)^2 - \rho^2} \begin{bmatrix} 1+\tau & -\rho \\ -\rho & 1+\tau \end{bmatrix} \otimes$ $I_d$. Furthermore,

$$(\Sigma + \tau I_p)^{-2} = \frac{1}{((1+\tau)^2 - \rho^2)^2} \begin{bmatrix} (1+\tau)^2 + \rho^2 & -2\rho(1+\tau) \\ -2\rho(1+\tau) & (1+\tau)^2 + \rho^2 \end{bmatrix} \otimes I_d,$$

$$\Sigma^2 = \begin{bmatrix} 1+\rho^2 & 2\rho \\ 2\rho & 1+\rho^2 \end{bmatrix} \otimes I_d.$$

A direct block multiplication gives

$$\mathrm{Tr}\left[\left(\Sigma(\Sigma + \tau I_p)^{-2}\right)_1\right] = d\, \frac{(1+\tau)^2 - \rho^2(1+2\tau)}{((1+\tau)^2 - \rho^2)^2},$$

$$\mathrm{Tr}\left[\Sigma^2(\Sigma + \tau I_p)^{-2}\right] = 2d\, \frac{(1+\tau-\rho^2)^2 + \rho^2\tau^2}{((1+\tau)^2 - \rho^2)^2},$$

$$\mathrm{Tr}\left[\left((\Sigma + \tau I_p)^{-2}\Sigma^2 D(\Sigma + \tau I_p)^{-1}\Sigma\right)_1\right] = \frac{(1+\tau-\rho^2)\left((1+\tau-\rho^2)^2 + \rho^2\tau^2\right)}{((1+\tau)^2 - \rho^2)^3} \mathrm{Tr}[\mathrm{diag}(b)]$$

$$+ \frac{2(1+\tau-\rho^2)\rho^2\tau^2}{((1+\tau)^2 - \rho^2)^3} \mathrm{Tr}[\mathrm{diag}(c)],$$

$$\mathrm{Tr}\left[\left((\Sigma + \tau I_p)^{-2}\Sigma D(\Sigma + \tau I_p)^{-1}\Sigma\right)_1\right] = \frac{(1+\tau-\rho^2)\left((1+\tau)(1+\tau-\rho^2) - \rho^2\tau\right)}{((1+\tau)^2 - \rho^2)^3} \mathrm{Tr}[\mathrm{diag}(b)]$$

$$+ \frac{\rho^2\tau\left(\rho^2 + \tau^2 - 1\right)}{((1+\tau)^2 - \rho^2)^3} \mathrm{Tr}[\mathrm{diag}(c)].$$

Expanding each rational function at $\rho = 0$ using

$$\frac{1}{((1+\tau)^2 - \rho^2)^2} = \frac{1}{(1+\tau)^4}\left(1 + \frac{2\rho^2}{(1+\tau)^2}\right) + O(\rho^4),$$

$$\frac{1}{((1+\tau)^2 - \rho^2)^3} = \frac{1}{(1+\tau)^6}\left(1 + \frac{3\rho^2}{(1+\tau)^2}\right) + O(\rho^4),$$

yields, to order $\rho^2$,

$$\frac{1}{d}\mathrm{Tr}\left[\left(\Sigma(\Sigma + \tau I_p)^{-2}\right)_1\right] = \left(\frac{1}{(1+\tau)^2} + \rho^2\frac{1-2\tau}{(1+\tau)^4}\right) + O(\rho^4),$$

$$\frac{1}{d}\mathrm{Tr}\left[\Sigma^2(\Sigma + \tau I_p)^{-2}\right] = \frac{2}{(1+\tau)^2} + 2\rho^2\frac{\tau^2 - 2\tau}{(1+\tau)^4} + O(\rho^4),$$

$$\frac{1}{d}\mathrm{Tr}\left[\left((\Sigma + \tau I_p)^{-2}\Sigma^2 D(\Sigma + \tau I_p)^{-1}\Sigma\right)_1\right] = \frac{\bar{b}}{(1+\tau)^3} + \rho^2\left(\frac{\tau^2 - 3\tau}{(1+\tau)^5}\bar{b} + \frac{2\tau^2}{(1+\tau)^5}\bar{c}\right) + O(\rho^4),$$

$$\frac{1}{d}\mathrm{Tr}\left[\left((\Sigma + \tau I_p)^{-2}\Sigma D(\Sigma + \tau I_p)^{-1}\Sigma\right)_1\right] = \frac{\bar{b}}{(1+\tau)^3} + \rho^2\left(\frac{1-3\tau}{(1+\tau)^5}\bar{b} + \frac{\tau(\tau^2-1)}{(1+\tau)^6}\bar{c}\right) + O(\rho^4).$$

Moreover, we have that

$$\frac{\kappa\,\mathrm{tr}\left[\Sigma^2(\Sigma + \tau I_p)^{-2}\right]}{p - \kappa\,\mathrm{tr}\left[\Sigma^2(\Sigma + \tau I_p)^{-2}\right]} = \frac{\kappa}{(1+\tau)^2 - \kappa} + \rho^2\frac{\kappa\tau(\tau - 2)}{((1+\tau)^2 - \kappa)^2} + O(\rho^4).$$

Plugging these into (57) gives the claimed result. $\qquad\square$

Let us further define

$$\tau^*(D, \rho) := \arg\min_{\tau \geq 0} \widetilde{\mathcal{R}}(D, \lambda, \rho), \qquad \tau_0^*(\rho) := \arg\min_{\tau \geq 0} \mathcal{R}_0(\lambda, \rho), \qquad \tau_0 := \tau_0^*(0). \qquad (58)$$

Then, the following result proves an expression for $\tau^*(D, \rho)$, up to order $\rho^2$.

**Lemma 13.** *In the setting of Theorem 4, we have that*

$$\tau^*(D, \rho) = \tau_0^*(\rho) + \bar{b}\left(B_3(\sigma, \kappa) + O(\rho^2)\right) + \bar{c}\left(\rho^2 C_3(\sigma, \kappa) + O(\rho^4)\right) + O(\bar{b}^2 + \bar{c}^2),$$

*where*

$$\tau_0 = \frac{1 + \kappa + \kappa\sigma^2 + \sqrt{(1 + \kappa + \kappa\sigma^2)^2 - 4\kappa}}{2} - 1,$$

$$\tau_0^*(\rho) = \tau_0 - \rho^2 \frac{\kappa\tau_0^2}{(1 + \tau_0)\left((1 + \tau_0)^2 - \kappa\right)} + O(\rho^4),$$

$$B_3(\sigma, \kappa) = -\frac{2(1 + \tau_0)^4 - 3(\kappa + 1)(1 + \tau_0)^3 + 4\kappa(1 + \tau_0)^2 + \kappa(\kappa + 1)(1 + \tau_0) - 2\kappa^2}{(1 + \tau_0)^2\left((1 + \tau_0)^2 - \kappa\right)},$$

$$C_3(\sigma, \kappa) = -\frac{\tau_0^2\left(4\tau_0^4 + (6 - 3\kappa)\tau_0^3 - (6 + 3\kappa)\tau_0^2 + (\kappa^2 + 9\kappa - 14)\tau_0 - 3\kappa^2 + 9\kappa - 6\right)}{(1 + \tau_0)^4\left((1 + \tau_0)^2 - \kappa\right)}.$$

(59)

*Proof.* A direct differentiation gives

$$\frac{\mathrm{d}}{\mathrm{d}\tau}\widetilde{\mathcal{R}}(D, \lambda, \rho) = \frac{2}{1 + \tau}\left(\frac{\tau}{(1 + \tau)^2 - \kappa} - \frac{\kappa\left(\sigma^2(1 + \tau)^2 + \tau^2\right)}{\left((1 + \tau)^2 - \kappa\right)^2}\right)$$

$$+ \rho^2\left(\frac{2\tau\left(\tau^2 - 4\tau + 1\right)}{(1 + \tau)^5} + \frac{2\kappa\tau\left((1 + \tau)^2\left(3\tau^2 - 5\tau + 1\right) - \kappa\left(\tau^2 - 4\tau + 1\right)\right)}{(1 + \tau)^5\left((1 + \tau)^2 - \kappa\right)^2}\right.$$

$$- \frac{2\kappa^2\left(\kappa\sigma^2(\tau - 1)(1 + \tau)^3 + \kappa\tau^2(\tau^2 + \tau - 3)\right)}{(1 + \tau)^3\left((1 + \tau)^2 - \kappa\right)^3}$$

$$\left.- \frac{2\kappa\left(\sigma^2(1 + \tau)^4(\tau^2 - 4\tau + 1) + \tau^2(1 + \tau)^2(\tau^2 - 5\tau + 3)\right)}{(1 + \tau)^3\left((1 + \tau)^2 - \kappa\right)^3}\right)$$

$$+ \bar{b}\left(\frac{4\tau - 2}{(1 + \tau)^4} + \frac{\kappa\tau(4 - 6\tau)}{(1 + \tau)^4\left((1 + \tau)^2 - \kappa\right)} - \frac{4\kappa^2\tau^2}{(1 + \tau)^4\left((1 + \tau)^2 - \kappa\right)^2}\right)$$

$$+ \bar{c}\rho^2\left(\frac{4\tau^2(2\tau - 3)}{(1 + \tau)^6} + \frac{2\kappa\tau^2(1 + \tau)\left(\kappa(\tau^2 - 6\tau + 3) - (1 + \tau)^2(3\tau^2 - 8\tau + 3)\right)}{(1 + \tau)^7\left((1 + \tau)^2 - \kappa\right)^2}\right).$$

With this explicit derivatives, the stationarity equation $\frac{\mathrm{d}}{\mathrm{d}\tau}\widetilde{\mathcal{R}}(D, \lambda, \rho) = 0$ is equivalent to $\frac{F\left(\tau, \bar{b}, \bar{c}, \rho^2\right)}{(1 + \tau)^7\left((1 + \tau)^2 - \kappa\right)^3} = 0$, where

$$F\left(\tau, \bar{b}, \bar{c}, \rho^2\right) = F_0(\tau)\rho^2 F_\rho(\tau) + \bar{b}F_b(\tau) + \bar{c}\rho^2 F_{\rho c}(\tau),$$

$$F_0(\tau) = 2(1 + \tau)^7\left(\kappa - (1 + \tau)^2\right)\left(\kappa\sigma^2\tau + \kappa\sigma^2 + \kappa\tau - \tau^2 - \tau\right),$$

$$F_b(\tau) = -2(1+\tau)^4\left(\kappa - (1+\tau)^2\right)\left(\kappa^2\tau - \kappa^2 - 3\kappa\tau^3 - 5\kappa\tau^2 + 2\kappa + 2\tau^4 + 5\tau^3 + 3\tau^2 - \tau - 1\right),$$

$$F_\rho(\tau) = -2(1 + \tau)^5\left(\kappa^2\sigma^2\tau^3 + \kappa^2\sigma^2\tau^2 - \kappa^2\sigma^2\tau - \kappa^2\sigma^2 + \kappa^2\tau^3 + \kappa^2\tau^2 - \kappa^2\tau\right.$$

$$+ \kappa\sigma^2\tau^5 - \kappa\sigma^2\tau^4 - 8\kappa\sigma^2\tau^3 - 8\kappa\sigma^2\tau^2 - \kappa\sigma^2\tau + \kappa\sigma^2$$

$$\left.+ \kappa\tau^5 - 4\kappa\tau^4 - 9\kappa\tau^3 - 2\kappa\tau^2 + 2\kappa\tau - \tau^6 + \tau^5 + 8\tau^4 + 8\tau^3 + \tau^2 - \tau\right).$$

$$F_{\rho c}(\tau) = -2\tau^2(1+\tau)^2\left(\kappa - (1+\tau)^2\right)\left(\kappa^2\tau - 3\kappa^2 - 3\kappa\tau^3 - 3\kappa\tau^2 + 9\kappa\tau + 9\kappa + 4\tau^4 + 6\tau^3 - 6\tau^2 - 14\tau - 6\right).$$

Setting $\bar{b} = \bar{c} = \rho^2 = 0$ yields

$$(1 + \tau)^2 - (1 + \kappa + \kappa\sigma^2)(1 + \tau) + \kappa = 0,$$

and the desired minimum corresponds to its largest solution, which is given by $\tau_0$ as expressed in the statement. It is easy to see that

$$\partial_\tau F\left(\tau_0, 0, 0, 0\right)) = -2(1 + \tau_0)^7\left((1 + \tau_0)^2 - \kappa\right)\left(2(1 + \tau_0) - (1 + \kappa + \kappa\sigma^2)\right)$$

$$= -2(1 + \tau_0)^7\left((1 + \tau_0)^2 - \kappa\right)\sqrt{(1 + \kappa + \kappa\sigma^2)^2 - 4\kappa} \neq 0.$$

Therefore, the implicit function theorem gives a smooth map $\tau^*(\bar{b}, \bar{c}, \rho^2)$ with $\tau^*(0,0,0) = \tau_0$ and $F(\tau^*, \cdot) = 0$. Differentiating $F = 0$ at $(\tau_0, 0, 0, 0)$ in each small parameter and dividing by $\partial_\tau F(\tau_0, 0, 0, 0)$ yields the linear expansion for $\tau^* - \tau_0$. The coefficient for $\rho^2$ in $\tau_0^*(\rho)$ is given by

$$\partial_{\rho^2} \tau^*(0,0,0) = -\left.\frac{\partial_{\rho^2} F}{\partial_\tau F}\right|_{(\tau_0,0,0,0)} = -\frac{F_\rho(\tau_0)}{\partial_\tau F_0(\tau_0, 0, 0, 0)}.$$

The $\bar{b}$ coefficient, $B_3$, is

$$B_3(\sigma, \kappa) = \partial_{\bar{b}} \tau^*(0,0,0) = -\left.\frac{\partial_{\bar{b}} F}{\partial_\tau F}\right|_{(\tau_0,0,0,0)} = -\frac{F_b(\tau_0)}{\partial_\tau F_0(\tau_0, 0, 0, 0)}.$$

The $\bar{c}\rho^2$ coefficient, $C_3$, is found from the mixed partial derivative:

$$C_3(\sigma, \kappa) = \partial_{\rho^2} \partial_{\bar{c}} \tau^*(0,0,0) = -\left.\frac{\partial_{\rho^2} \partial_{\bar{c}} F}{\partial_\tau F}\right|_{(\tau_0,0,0,0)} = -\frac{F_{\rho c}(\tau_0)}{\partial_\tau F_0(\tau_0, 0, 0, 0)}.$$

Substituting the expressions for $\partial_\tau F_0(\tau_0)$, $F_\rho(\tau_0)$, $F_b(\tau_0)$, and $F_{\rho c}(\tau_0)$ and cancelling common factors gives the coefficients as stated in (59). $\qquad\square$

As $\lambda$ and $\tau$ are linked by the fixed point equation (17), an application of Lemma 13 readily gives that

$$\lambda_{\text{eq}}^*(D, \rho) = \lambda_{\text{eq}, D=0}^*(\rho) + \bar{b}(B_1(\sigma, \kappa) + O(\rho^2)) + \bar{c}\rho^2(C_1(\sigma, \kappa) + O(\rho^2)) + O(\bar{b}^2 + \bar{c}^2), \quad (60)$$

where

$$\lambda_{\text{eq}, D=0}^*(\rho) = \tau_0\left(\kappa^{-1} - \frac{1}{1+\tau_0}\right) + \rho^2\left(\tau_0\left(\frac{1}{(1+\tau_0)^2} - \frac{1}{(1+\tau_0)^3}\right)\right.$$
$$\left. - \left(\kappa^{-1} - \frac{1}{(1+\tau_0)^2}\right)\frac{\kappa\tau_0^2}{(1+\tau_0)\left((1+\tau_0)^2 - \kappa\right)}\right) + O(\rho^4),$$

$$B_1(\sigma, \kappa) = -\frac{2(1+\tau_0)^4 - 3(\kappa+1)(1+\tau_0)^3 + 4\kappa(1+\tau_0)^2 + \kappa(\kappa+1)(1+\tau_0) - 2\kappa^2}{\kappa(1+\tau_0)^4},$$

$$C_1(\sigma, \kappa) = -\frac{\tau_0^2\left(4\tau_0^4 + (6 - 3\kappa)\tau_0^3 - (6 + 3\kappa)\tau_0^2 + (\kappa^2 + 9\kappa - 14)\tau_0 - 3\kappa^2 + 9\kappa - 6\right)}{\kappa(1+\tau_0)^6}.$$
$$(61)$$

This proves (22). Next, the corollary below proves (23).

**Corollary 14.** *Consider the setting of Theorem 4 and let $\tau_0$ be given by* (59). *Then, we have that*

$$\mathcal{R}_{\text{eq}}^*(D, \rho) = \mathcal{R}_{\text{eq}}^*(\rho) + \bar{b}(B_2(\sigma, \kappa) + O(\rho^2)) + \bar{c}\rho^2(C_2(\sigma, \kappa) + O(\rho^2)) + O(\bar{b}^2 + \bar{c}^2), \quad (62)$$

*where*

$$\mathcal{R}_{\text{eq}}^*(\rho) = \frac{\tau_0^2}{(1+\tau_0)^2} + \frac{\kappa}{(1+\tau_0)^2 - \kappa}\left(\sigma^2 + \frac{\tau_0^2}{(1+\tau_0)^2}\right)$$
$$+ \rho^2\left(\frac{\tau_0^2(1 - 2\tau_0)}{(1+\tau_0)^4} + \frac{\kappa\tau_0^2(1 - 2\tau_0)}{(1+\tau_0)^4\left((1+\tau_0)^2 - \kappa\right)}\right.$$
$$\left. + \frac{\kappa\tau_0(\tau_0 - 2)}{\left((1+\tau_0)^2 - \kappa\right)^2}\left(\sigma^2 + \frac{\tau_0^2}{(1+\tau_0)^2}\right)\right) + O(\rho^4), \quad (63)$$

$$B_2(\sigma, \kappa) = -\frac{2\tau_0}{(1+\tau_0)^3} + \frac{2\kappa\tau_0^2}{(1+\tau_0)^3\left((1+\tau_0)^2 - \kappa\right)},$$

$$C_2(\sigma, \kappa) = -\frac{4\tau_0^3}{(1+\tau_0)^5} + \frac{2\kappa\tau_0^3(\tau_0^2 - 1)}{(1+\tau_0)^6\left((1+\tau_0)^2 - \kappa\right)}.$$

*Proof.* Let us re-define $\widetilde{\mathcal{R}}(D, \lambda, \rho)$ given in (56) as $\widetilde{R}(\tau, \bar{b}, \bar{c}, \rho^2)$ to emphasize its dependence on $\tau, \bar{b}, \bar{c}$. By definition of $\tau_0$, we have $\partial_\tau \widetilde{R}(\tau_0, 0, 0, 0) = 0$. Furthermore, from Lemma 13, we have

$$\tau^*(D, \rho) = \tau_0 + O(\bar{b} + (1 + \bar{c})\rho^2).$$

A first–order Taylor expansion of $\widetilde{R}(\tau^*(D, \rho), \bar{b}, \bar{c}, \rho^2)$ around $(\tau; \bar{b}, \bar{c}, \rho^2) = (\tau_0; 0, 0, 0)$ gives

$$\widetilde{R}(\tau^*(D, \rho), \bar{b}, \bar{c}, \rho^2) = \widetilde{R}(\tau_0, \bar{b}, \bar{c}, \rho^2) + \partial_\tau \widetilde{R}(\tau_0, 0, 0, 0)(\tau^*(D, \rho) - \tau_0)$$
$$+ O\left((\tau^*(D, \rho) - \tau_0)\bar{b}\right) + O\left((\tau^*(D, \rho) - \tau_0)\rho^2\right) + O(\bar{b}^2 + \bar{c}^2 + \rho^4).$$

As $\partial_\tau \widetilde{R}(\tau_0, 0, 0, 0) = 0$, we conclude that

$$\widetilde{R}(\tau^*, \bar{b}, \bar{c}, \rho^2) = \widetilde{R}(\tau_0, \bar{b}, \bar{c}, \rho^2) + O(\bar{b}^2 + \bar{c}^2 + \rho^4),$$

and substituting $\tau = \tau_0$ in (56) gives the claimed expansion. $\square$

We now move to the proof of (24), which follows from the lemma below.

**Lemma 15.** *Let $B_1(\sigma, \kappa)$ be given by (61). Then, for any $\kappa > 1$, $B_1(\kappa, \cdot)$ has exactly one zero $\sigma_{B_1}(\kappa) > 0$, with*

$$B_1(\kappa, \sigma) \geq 0 \text{ for } 0 \leq \sigma \leq \sigma_{B_1}(\kappa), \qquad B_1(\kappa, \sigma) \leq 0 \text{ for } \sigma \geq \sigma_{B_1}(\kappa).$$

*Moreover, as $\kappa \to \infty$,*

$$\sigma_{B_1}^2(\kappa) = \frac{1}{2} - \frac{7}{18}\kappa^{-1} + O\left(\kappa^{-2}\right).$$

*Proof.* Let us define the shorthands

$$s(\sigma) := 1 + \tau_0, \qquad N_{B_1}(s, \kappa) := 2s^4 - 3(\kappa + 1)s^3 + 4\kappa s^2 + \kappa(\kappa + 1)s - 2\kappa^2, \qquad (64)$$

with $\tau_0$ given by (59). Note that

$$s(\sigma) = \frac{1 + \kappa + \kappa\sigma^2 + \sqrt{(1 + \kappa + \kappa\sigma^2)^2 - 4\kappa}}{2} \geq \frac{1 + \kappa + \sqrt{(1 + \kappa)^2 - 4\kappa}}{2} = \kappa.$$

Now let us also define $\Phi(s) := -N_{B_1}(s, \kappa)/\left(\kappa s^4\right)$ for $s \geq \kappa$. A direct calculation gives the factorization

$$\frac{\mathrm{d}}{\mathrm{d}s}\Phi(s) = \frac{-N_{B_1}'(s)s + 4N_{B_1}(s)}{\kappa s^5} = \frac{-(s^2 - \kappa)(3(\kappa + 1)s - 8\kappa)}{\kappa s^5}.$$

For $s \geq \kappa$ we have $s^2 - \kappa > 0$, hence $\Phi'(s)$ changes sign only once at $s_* := \frac{8\kappa}{3(\kappa+1)}$. If $\kappa \geq 5/3$, then $s_* \leq \kappa$ and $\Phi$ is strictly decreasing on $[\kappa, \infty)$. If $1 < \kappa < 5/3$, then $\kappa < s_*$ and $\Phi$ is increasing on $[\kappa, s_*)$ and strictly decreasing on $(s_*, \infty)$.

Note that $B_1(\kappa, \sigma) = -N_{B_1}(s(\sigma), \kappa)/(\kappa s(\sigma)^4) = \Phi(s(\sigma))$, $s(\sigma)$ is strictly increasing in $\sigma$, and $\Phi(s(\sigma)) \to -2/\kappa$ as $\sigma \to \infty$ (since $s(\sigma) \to \kappa\sigma^2$ and $N_{B_1}(s, \kappa) \to 2s^4$). Furthermore, $s(0) = \kappa$ and

$$N_{B_1}(\kappa, \kappa) = -\kappa^2(\kappa - 1)^2 < 0 \implies B_1(\kappa, 0) = -\frac{N_{B_1}(\kappa, \kappa)}{\kappa^5} > 0.$$

Therefore, $B_1(\kappa, \sigma)$ is strictly decreasing on $[0, \infty)$ if $\kappa \geq 5/3$, and for $1 < \kappa < 5/3$ it increases for small $\sigma$ and then strictly decreases; in either case, since $B_1(\kappa, 0) > 0$ and $\lim_{\sigma \to \infty} B_1(\kappa, \sigma) = -2/\kappa < 0$, it crosses 0 exactly once, which proves the existence and uniqueness of $\sigma_{B_1}(\kappa)$. At the crossing $B_1(\kappa, \sigma_{B_1}) = 0$, hence $N_{B_1}(s(\sigma_{B_1}), \kappa) = 0$.

Now let $\varepsilon := \kappa^{-1}$ and write $s = \kappa c$. Dividing $N_{B_1}(\kappa c, \kappa) = 0$ by $\kappa^4$ yields the analytic equation

$$F(\varepsilon, c) = 0, \qquad F(\varepsilon, c) := 2c^4 - 3(1 + \varepsilon)c^3 + 4\varepsilon c^2 + (\varepsilon + \varepsilon^2)c - 2\varepsilon^2.$$

At $\varepsilon = 0$, $F(0, c) = 2c^4 - 3c^3$ has the positive root $c_0 = \frac{3}{2}$, and $\partial_c F(0, c_0) = 8c_0^3 - 9c_0^2 = \frac{27}{4} \neq 0$. By the implicit function theorem there exists a unique analytic branch $c(\varepsilon)$ with $c(0) = \frac{3}{2}$, having the expansion $c(\varepsilon) = \frac{3}{2} + c_1\varepsilon + O(\varepsilon^2)$. Substituting into $F(\varepsilon, c) = 0$ gives that, up to first order,

$$\frac{27}{4}c_1 + \frac{3}{8} = 0 \implies c_1 = -\frac{1}{18}.$$

Thus,

$$\sigma_{B_1}^2 = \frac{(s_c - 1)(s_c - \kappa)}{\kappa s_c} = \left(1 - \frac{1}{\kappa c(\varepsilon)}\right)(c(\varepsilon) - 1),$$

with $s_c = \kappa c(\varepsilon) = \frac{3}{2}\kappa - \frac{1}{18} + O(\kappa^{-1})$. Substituting $c(\varepsilon) = \frac{3}{2} - \frac{1}{18}\varepsilon + O(\varepsilon^2)$ and expanding yields

$$\sigma_{B_1}^2 = \frac{1}{2} - \frac{7}{18}\kappa^{-1} + O(\kappa^{-2}).$$

The analyticity of $c(\varepsilon)$ implies the remainder $O(\varepsilon^2)$ in $c$ and, consequently, the remainder $O(\kappa^{-2})$ in the displayed expansion. $\qquad\square$

Next, we move to the proof of (25), which follows from the lemma below.

**Lemma 16.** *Let $C_1(\sigma, \kappa)$ be given by* (61). *Then, for every $\kappa \geq 2$ and all $\sigma \geq 0$, $C_1(\kappa, \sigma) \leq 0$.*

*Proof.* Let $s(\sigma) = 1 + \tau_0$, with $\tau_0$ given by (59), and note that $\tau_0^2/(\kappa s(\sigma)^6) > 0$. Thus, the sign of $C_1$ is the opposite of the sign of $N_{C_1}(s(\sigma) - 1, \kappa)$, where

$$N_{C_1}(t, \kappa) = 4t^4 + (6 - 3\kappa)t^3 - (6 + 3\kappa)t^2 + (\kappa^2 + 9\kappa - 14)t - 3\kappa^2 + 9\kappa - 6.$$

At $\sigma = 0$, one has $s(0) - 1 = \kappa - 1$, and a direct substitution yields

$$N_{C_1}(\kappa - 1, \kappa) := \kappa^4 - 3\kappa^3 + 2\kappa^2 = \kappa^2(\kappa - 1)(\kappa - 2).$$

Moreover, differentiating in $t$ gives

$$N_{C_1}''(t, \kappa) = 6(-3\kappa t - \kappa + 8t^2 + 6t - 2) > 0,$$

for all $t \geq \kappa - 1$ and $\kappa \geq 2$, so $N_{C_1}'(\cdot, \kappa)$ is increasing on $[\kappa - 1, \infty)$. As

$$N_{C_1}'(\kappa - 1, \kappa) = \kappa(\kappa - 2)(7\kappa - 3) \geq 0,$$

for $\kappa \geq 2$, it follows that $N_{C_1}(\cdot, \kappa)$ is increasing on $[\kappa - 1, \infty)$. Therefore, for every $\sigma \geq 0$,

$$N_{C_1}(s(\sigma) - 1, \kappa) \geq N_{C_1}(\kappa - 1, \kappa) = \kappa^2(\kappa - 1)(\kappa - 2) \geq 0 \quad \text{for } \kappa \geq 2.$$

Since the prefactor is positive, $C_1(\kappa, \sigma) \leq 0$ for all $\sigma$ as soon as $\kappa \geq 2$. $\qquad\square$

**Lemma 17.** *Let $B_2(\sigma, \kappa)$ be given by* (63). *Then, for every $\kappa > 1$ and all $\sigma \geq 0$, $B_2(\kappa, \sigma) \leq 0$.*

*Proof.* Recall that

$$B_2(\sigma, \kappa) = -\frac{2\tau_0}{(1 + \tau_0)^3} + \frac{2\kappa\tau_0^2}{(1 + \tau_0)^3\left((1 + \tau_0)^2 - \kappa\right)} = \frac{2\tau_0\left(\kappa - (1 + \tau_0)\right)}{(1 + \tau_0)^2\left((1 + \tau_0)^2 - \kappa\right)}.$$

Since $(1 + \tau_0)^2 > (1 + \tau_0) \geq \kappa$, we have $B_2(\sigma, \kappa) \leq 0$. $\qquad\square$

**Lemma 18.** *Let $C_2(\sigma, \kappa)$ be given by* (63). *Then, for every $\kappa > 1$ and all $\sigma \geq 0$, $C_2(\kappa, \sigma) \leq 0$.*

*Proof.* Let us again write $s(\sigma) = 1 + \tau_0 > 0$ and note $\tau_0^2 - 1 = (s(\sigma) - 1)^2 - 1 = s(\sigma)^2 - 2s(\sigma)$. Then, substituting and simplifying, we have

$$C_2(\kappa, \sigma) = -\frac{4\tau_0^3}{s(\sigma)^5} + \frac{2\tau_0^3}{s(\sigma)^6}\frac{\kappa(s(\sigma)^2 - 2s(\sigma))}{s(\sigma)^2 - \kappa} = \frac{2\tau_0^3}{s(\sigma)^6}\left(\frac{\kappa s(\sigma)^2 - 2s(\sigma)^3}{s(\sigma)^2 - \kappa}\right) = \frac{2\tau_0^3}{s(\sigma)^4}\frac{\kappa - 2s(\sigma)}{s(\sigma)^2 - \kappa}.$$

Note that $s(\sigma) > \kappa$, and therefore $s(\sigma)^2 - \kappa > 0$ and $\kappa - 2s(\sigma) \leq \kappa - 2\kappa = -\kappa < 0$. Because $\tau_0 > 0$ for $\kappa > 1$, the prefactor $2\tau_0^3/s(\sigma)^4 > 0$. Therefore $C_2(\kappa, \sigma) \leq 0$. $\qquad\square$

Combining the results from Lemmas 15, 16, 17 and 18 concludes the proof of Theorem 4.

## D   TEST RISK EVALUATED ON $\mathcal{D}(\theta)$ IN THE OVER-PARAMETERIZED SETTING

Let $\overline{\mathcal{R}}_k(\Sigma, \theta_k, \theta_{\text{pop}}^*)$ be the excess risk of the estimator $\theta_k$ given by (14) evaluated on $\mathcal{D}(\theta)$, i.e.,

$$\overline{\mathcal{R}}_k(\Sigma, \theta_k, \theta_{\text{pop}}^*) = \left\| \theta_k - (\theta_{\text{pop}}^* + D\theta_{k-1}) \right\|_{\Sigma}^2.$$

**Theorem 19** (Excess risk on $\mathcal{D}(\theta)$– over-parameterized). *Let $R > 0$ be a constant s.t. $\theta_{\text{pop}}^*, \theta_0 \in B_p(R)$. Assume that $\kappa, \sigma, \lambda \in (1/M, M)$ and $\|\Sigma\|_{\text{op}}, \|\Sigma^{-1}\|_{\text{op}} \leq M$ for some constant $M > 1$. Then, there exists a constant $C = C(M, R)$ such that for any $\delta \in (0, 1/2]$, with probability at least $1 - Cpe^{-p\delta^4/C}$,*

$$\left| \overline{\mathcal{R}}_2(\Sigma, \theta_2, \theta_{\text{pop}}^*) - \overline{\mathcal{R}}_{\text{eq}}\left(\Sigma, \theta_{\text{pop}}^*, D, \lambda\right) \right| \leq \delta + O(\|D\|_{\text{op}}^2), \tag{65}$$

*where*

$$\overline{\mathcal{R}}_{\text{eq}}\left(\Sigma, \theta_{\text{pop}}^*, D, \lambda\right)$$
$$= \frac{\sigma^2 \kappa \operatorname{Tr}\left[\Sigma^2 \left(\Sigma + \tau I_p\right)^{-2}\right] + p\tau^2 \langle \theta_{\text{pop}}^*, \left(\Sigma + \tau I_p\right)^{-1}\left(I_p + 2\left(\Sigma + \tau I_p\right)^{-1}\Sigma D\right)\Sigma\left(\Sigma + \tau I_p\right)^{-1}\theta_{\text{pop}}^* \rangle}{p - \kappa \operatorname{Tr}\left[\Sigma^2 \left(\Sigma + \tau I_p\right)^{-2}\right]},$$
$$\tag{66}$$

*and $\tau$ is the unique solution of (17).*

*Proof.* The argument is analogous to that used to prove Theorem 3, and we only report differences. Using the same approach as Lemma 7, we have

$$\sup_{\theta_{\text{pop}}^*, \theta_0 \in B_p(R)} \Pr\left(\left| \overline{\mathcal{R}}_1(\Sigma, \theta_1, \theta_{\text{pop}}^*) - \overline{\mathcal{R}}_{\text{eq}}^{(1)}\left(\Sigma, \theta_0, \theta_{\text{pop}}^*\right) \right| \geq \delta\right) \leq Cpe^{-p\delta^4/C}, \tag{67}$$

where

$$\overline{\mathcal{R}}_{\text{eq}}^{(1)}\left(\Sigma, \theta_0, \theta_{\text{pop}}^*\right) = \frac{\sigma^2 \kappa \operatorname{Tr}\left[\Sigma^2 \left(\Sigma + \tau I_p\right)^{-2}\right] + p\tau^2 \left\| \left(\Sigma + \tau I_p\right)^{-1}\left(\theta_{\text{pop}}^* + D\theta_0\right) \right\|_{\Sigma}^2}{p - \kappa \operatorname{Tr}\left[\Sigma^2 \left(\Sigma + \tau I_p\right)^{-2}\right]}. \tag{68}$$

Next, using the same approach[3] as Lemma 8, we have

$$\sup_{\theta_{\text{pop}}^*, \theta_0 \in B_p(R)} \Pr\left(\left| \overline{\mathcal{R}}_2(\Sigma, \theta_2, \theta_{\text{pop}}^*) - \overline{\mathcal{R}}_{\text{eq}}^{(2)}\left(\Sigma, \theta_0, \theta_{\text{pop}}^*\right) \right| \geq \delta\right) \leq Cpe^{-p\delta^4/C}, \tag{69}$$

where

$$\overline{\mathcal{R}}_{\text{eq}}^{(2)}\left(\Sigma, \theta_0, \theta_{\text{pop}}^*\right)$$
$$= \frac{\sigma^2 \kappa \operatorname{Tr}\left[\Sigma^2 \left(\Sigma + \tau I_p\right)^{-2}\right] + p\tau^2 \left\| \left(\Sigma + \tau I_p\right)^{-1}\left(\theta_{\text{pop}}^* + D\left(\Sigma + \tau I_p\right)^{-1}\Sigma(\theta_{\text{pop}}^* + D\theta_0)\right) \right\|_{\Sigma}^2}{p - \kappa \operatorname{Tr}\left[\Sigma^2 \left(\Sigma + \tau I_p\right)^{-2}\right]}$$
$$+ p\kappa\tau^2 \operatorname{Tr}\left[\Sigma\left(\Sigma + \tau I_p\right)^{-2} D\Sigma\left(\Sigma + \tau I_p\right)^{-2} D\right] \cdot \frac{\sigma^2 + \tau^2 \left\| \left(\Sigma + \tau I_p\right)^{-1}\left(\theta_{\text{pop}}^* + D\theta_0\right) \right\|_{\Sigma}^2}{\left(p - \kappa \operatorname{Tr}\left[\Sigma^2 \left(\Sigma + \tau I_p\right)^{-2}\right]\right)^2}.$$
$$\tag{70}$$

Noting that the quantity in the second line is $O(\|D\|_{\text{op}}^2)$ and that

$$\left\| \left(\Sigma + \tau I_p\right)^{-1}\left(\theta_{\text{pop}}^* + D\left(\Sigma + \tau I_p\right)^{-1}\Sigma(\theta_{\text{pop}}^* + D\theta_0)\right) \right\|_{\Sigma}^2$$
$$= \langle \theta_{\text{pop}}^*, \left(\Sigma + \tau I_p\right)^{-1}\left(I_p + 2\left(\Sigma + \tau I_p\right)^{-1}\Sigma D\right)\Sigma\left(\Sigma + \tau I_p\right)^{-1}\theta_{\text{pop}}^* \rangle + O(\|D\|_{\text{op}}^2) \tag{71}$$

concludes the argument. □

---

[3]In fact, the derivation is simpler since the term corresponding to $\mathcal{R}_{\text{eq},1}^{(1)}\left(\Sigma, \theta_1, \theta_{\text{pop}}^*\right)$ here is absent.

**Lemma 20.** *Consider the setting of Theorem 19, assume that $a$ has covariance $I_d/d$, and let $\Sigma = \begin{bmatrix} I_d & \rho I_d \\ \rho I_d & I_d \end{bmatrix}$. Then, we have that*

$$\mathbb{E}_{\theta^*_{\text{pop}}} \overline{\mathcal{R}}_{\text{eq}} \left( \Sigma, \theta^*_{\text{pop}}, D, \lambda \right) = \overline{\mathcal{R}}(D, \lambda, \rho) + O(\bar{b}\rho^2 + \rho^4),$$

$$\overline{\mathcal{R}}(D, \lambda, \rho) := \mathcal{R}_0(\lambda, \rho) + \bar{b}\overline{A}_1(\lambda) + \bar{c}\rho^2 \overline{A}_2(\lambda),$$

*where $\bar{b} = \text{Tr}[\text{diag}(b)]/d, \bar{c} = \text{Tr}[\text{diag}(c)]/d$, the auxiliary function $\mathcal{R}_0(\lambda, \rho)$ is given by (56), and the new auxiliary functions $\overline{A}_1(\lambda)$ and $\overline{A}_2(\lambda)$ are given by*

$$\overline{A}_1(\lambda) = \frac{2\tau^2}{(1+\tau)((1+\tau)^2 - \kappa)},$$

$$\overline{A}_2(\lambda) = \frac{2\tau^3(\tau^2 - 1)}{(1+\tau)^4((1+\tau)^2 - \kappa)}. \tag{72}$$

*Proof.* Using $\mathbb{E}_{\theta^*_{\text{pop}}} \left[ \langle \theta^*_{\text{pop}}, M\theta^*_{\text{pop}} \rangle \right] = \text{Tr}\left[ (M)_1 \right]/d$ from the proof of Lemma 12, we take the expectation of (66):

$$\mathbb{E}_{\theta^*_{\text{pop}}} \overline{\mathcal{R}}_{\text{eq}} \left( \Sigma, \theta^*_{\text{pop}}, D, \lambda \right)$$

$$= \frac{\sigma^2 \kappa \, \text{Tr}\left[ \Sigma^2 \left( \Sigma + \tau I_p \right)^{-2} \right]}{p - \kappa \, \text{Tr}\left[ \Sigma^2 \left( \Sigma + \tau I_p \right)^{-2} \right]}$$

$$+ \frac{p\tau^2}{p - \kappa \, \text{Tr}\left[ \Sigma^2 \left( \Sigma + \tau I_p \right)^{-2} \right]} \frac{1}{d} \text{Tr}\left[ \left( \Sigma \left( \Sigma + \tau I_p \right)^{-2} \right)_1 \right] \tag{73}$$

$$+ \frac{p\tau^2}{p - \kappa \, \text{Tr}\left[ \Sigma^2 \left( \Sigma + \tau I_p \right)^{-2} \right]} \frac{2}{d} \text{Tr}\left[ \left( \left( \Sigma + \tau I_p \right)^{-2} \Sigma D \left( \Sigma + \tau I_p \right)^{-1} \Sigma \right)_1 \right].$$

The first two terms correspond to the risk with $D = 0$ (i.e., $\bar{b} = \bar{c} = 0$). By the same computations as in the proof of Lemma 12, these terms combine to $\mathcal{R}_0(\lambda, \rho) + O(\rho^4)$. The third term, which depends on $D$, requires approximations for its two factors. The first factor is new:

$$\frac{p}{p - \kappa \, \text{Tr}\left[ \Sigma^2 (\Sigma + \tau I_p)^{-2} \right]} = 1 + \frac{\kappa \, \text{tr}\left[ \Sigma^2 (\Sigma + \tau I_p)^{-2} \right]}{p - \kappa \, \text{tr}\left[ \Sigma^2 (\Sigma + \tau I_p)^{-2} \right]}$$

$$= 1 + \left( \frac{\kappa}{(1+\tau)^2 - \kappa} + O(\rho^2) \right) = \frac{(1+\tau)^2}{(1+\tau)^2 - \kappa} + O(\rho^2).$$

For the second factor, we use the trace expansion from Lemma 12:

$$\frac{1}{d} \text{Tr}\left[ \left( (\Sigma + \tau I_p)^{-2} \Sigma D (\Sigma + \tau I_p)^{-1} \Sigma \right)_1 \right] = \frac{\bar{b}}{(1+\tau)^3} + \rho^2 \left( \frac{1 - 3\tau}{(1+\tau)^5} \bar{b} + \frac{\tau(\tau^2 - 1)}{(1+\tau)^6} \bar{c} \right) + O(\rho^4).$$

We multiply these two factors by $2\tau^2$ (from (73)) and keep only the terms of order $O(\bar{b})$ and $O(\bar{c}\rho^2)$:

$$\bar{b}\overline{A}_1(\lambda) = \left( \frac{(1+\tau)^2}{(1+\tau)^2 - \kappa} \right) \left( 2\tau^2 \frac{\bar{b}}{(1+\tau)^3} \right) = \bar{b} \frac{2\tau^2}{(1+\tau)((1+\tau)^2 - \kappa)},$$

$$\bar{c}\rho^2 \overline{A}_2(\lambda) = \left( \frac{(1+\tau)^2}{(1+\tau)^2 - \kappa} \right) \left( 2\tau^2 \rho^2 \bar{c} \frac{\tau(\tau^2 - 1)}{(1+\tau)^6} \right) = \bar{c}\rho^2 \frac{2\tau^3(\tau^2 - 1)}{(1+\tau)^4((1+\tau)^2 - \kappa)}.$$

Adding these terms to $\mathcal{R}_0(\lambda, \rho)$ yields the claimed expansion. □

**Lemma 21.** *In the setting of Lemma 20, we have that*

$$\tau^*(D,\rho) = \tau_0^*(\rho) + \bar{b}\left(\overline{B}_3(\sigma,\kappa) + O(\rho^2)\right) + \bar{c}\left(\rho^2\overline{C}_3(\sigma,\kappa) + O(\rho^4)\right) + O(\bar{b}^2 + \bar{c}^2),$$

*where $\tau_0$ and $\tau_0^*(\rho)$ are given by (59), and*

$$\overline{B}_3(\sigma,\kappa) = -\frac{\tau_0\left((1+\tau_0)^2(2-\tau_0) - \kappa(\tau_0+2)\right)}{(1+\tau_0)\left((1+\tau_0)^2 - \kappa\right)},$$

$$\overline{C}_3(\sigma,\kappa) = -\frac{\tau_0^2\left((4\tau_0-3)(1+\tau_0)((1+\tau_0)^2 - \kappa) - \tau_0(\tau_0-1)(5(1+\tau_0)^2 - 3\kappa)\right)}{(1+\tau_0)^3\left((1+\tau_0)^2 - \kappa\right)}.$$

(74)

*Proof.* A direct differentiation of $\overline{\mathcal{R}}(D,\lambda,\rho)$ from Lemma 20 gives

$$\frac{\mathrm{d}}{\mathrm{d}\tau}\overline{\mathcal{R}}(D,\lambda,\rho) = \frac{\mathrm{d}}{\mathrm{d}\tau}\mathcal{R}_0(\lambda,\rho) + \bar{b}\frac{\mathrm{d}}{\mathrm{d}\tau}\overline{A}_1(\lambda) + \bar{c}\rho^2\frac{\mathrm{d}}{\mathrm{d}\tau}\overline{A}_2(\lambda).$$

The first term is identical to that in the proof of Lemma 13. The new derivatives are:

$$\frac{\mathrm{d}}{\mathrm{d}\tau}\overline{A}_1(\lambda) = \frac{2\tau(1+\tau)^2(2-\tau) - 2\kappa\tau(2+\tau)}{(1+\tau)^2((1+\tau)^2 - \kappa)^2},$$

$$\frac{\mathrm{d}}{\mathrm{d}\tau}\overline{A}_2(\lambda) = \frac{2\tau^2\left((4\tau-3)(1+\tau)((1+\tau)^2 - \kappa) - \tau(\tau-1)(5(1+\tau)^2 - 3\kappa)\right)}{(1+\tau)^4((1+\tau)^2 - \kappa)^2}.$$

With these explicit derivatives, the stationarity equation $\frac{\mathrm{d}}{\mathrm{d}\tau}\overline{\mathcal{R}}(D,\lambda,\rho) = 0$ is equivalent to $\frac{\overline{F}\left(\tau,\bar{b},\bar{c},\rho^2\right)}{(1+\tau)^7((1+\tau)^2 - \kappa)^3} = 0$, where

$$\overline{F}\left(\tau,\bar{b},\bar{c},\rho^2\right) = F_0(\tau) + \rho^2 F_\rho(\tau) + \bar{b}\overline{F}_b(\tau) + \bar{c}\rho^2\overline{F}_{\rho c}(\tau).$$

The functions $F_0(\tau)$ and $F_\rho(\tau)$ are identical to those defined in the proof of Lemma 13. The new functions are

$$\overline{F}_b(\tau) = 2\tau(1+\tau)^5((1+\tau)^2 - \kappa)\left((1+\tau)^2(2-\tau) - \kappa(2+\tau)\right),$$

$$\overline{F}_{\rho c}(\tau) = 2\tau^2(1+\tau)^3((1+\tau)^2 - \kappa)\left((4\tau-3)(1+\tau)((1+\tau)^2 - \kappa) - \tau(\tau-1)(5(1+\tau)^2 - 3\kappa)\right).$$

Setting $\bar{b} = \bar{c} = \rho^2 = 0$ yields the same equation for $\tau_0$ as in Lemma 13. The partial derivative $\partial_\tau\overline{F}\left(\tau_0,0,0,0\right)$ is also unchanged:

$$\partial_\tau\overline{F}\left(\tau_0,0,0,0)\right) = -2(1+\tau_0)^7\left((1+\tau_0)^2 - \kappa\right)\sqrt{(1+\kappa+\kappa\sigma^2)^2 - 4\kappa} \neq 0.$$

Therefore, the implicit function theorem gives a smooth map $\tau^*(\bar{b},\bar{c},\rho^2)$ with $\tau^*(0,0,0) = \tau_0$ and $\overline{F}(\tau^*,\cdot) = 0$. Differentiating $\overline{F} = 0$ at $(\tau_0,0,0,0)$ and dividing by $\partial_\tau\overline{F}(\tau_0,0,0,0)$ yields

$$\overline{B}_3(\sigma,\kappa) = -\frac{\overline{F}_b(\tau_0)}{\partial_\tau F(\tau_0,0,0,0)}, \qquad \overline{C}_3(\sigma,\kappa) = -\frac{\overline{F}_{\rho c}(\tau_0)}{\partial_\tau F(\tau_0,0,0,0)}.$$

Substituting the expressions for $\overline{F}_b(\tau_0)$, $\overline{F}_{\rho c}(\tau_0)$, and $\partial_\tau F(\tau_0,0,0,0)$ and cancelling common factors gives the coefficients as stated in (74). □

As $\lambda$ and $\tau$ are linked by the fixed point equation (17), an application of Lemma 21 readily gives that

$$\lambda_{\mathrm{eq}}^*(D,\rho) = \lambda_{\mathrm{eq},D=0}^*(\rho) + \bar{b}(\overline{B}_1(\sigma,\kappa) + O(\rho^2)) + \bar{c}\rho^2(\overline{C}_1(\sigma,\kappa) + O(\rho^2)) + O(\bar{b}^2 + \bar{c}^2), \quad (75)$$

where $\lambda_{\mathrm{eq},D=0}^*(\rho)$ is unchanged from (61), and the new coefficients are

$$\overline{B}_1(\sigma,\kappa) = -\frac{\tau_0\left((1+\tau_0)^2(2-\tau_0) - \kappa(\tau_0+2)\right)}{\kappa(1+\tau_0)^3},$$

$$\overline{C}_1(\sigma,\kappa) = -\frac{\tau_0^2\left((4\tau_0-3)(1+\tau_0)((1+\tau_0)^2 - \kappa) - \tau_0(\tau_0-1)(5(1+\tau_0)^2 - 3\kappa)\right)}{\kappa(1+\tau_0)^5}.$$

(76)

Next, the corollary below provides the expansion for the optimal equilibrium risk.

**Corollary 22.** *Consider the setting of Lemma 20 and let $\tau_0$ be given by (59). Then, we have that*

$$\overline{\mathcal{R}}_{\mathrm{eq}}^*(D, \rho) = \mathcal{R}_{\mathrm{eq}}^*(\rho) + \bar{b}(\overline{B}_2(\sigma, \kappa) + O(\rho^2)) + \bar{c}\rho^2(\overline{C}_2(\sigma, \kappa) + O(\rho^2)) + O(\bar{b}^2 + \bar{c}^2), \quad (77)$$

*where $\mathcal{R}_{\mathrm{eq}}^*(\rho)$ is given by (63), and*

$$\begin{aligned}
\overline{B}_2(\sigma, \kappa) &= \frac{2\tau_0^2}{(1 + \tau_0)((1 + \tau_0)^2 - \kappa)}, \\
\overline{C}_2(\sigma, \kappa) &= \frac{2\tau_0^3(\tau_0^2 - 1)}{(1 + \tau_0)^4((1 + \tau_0)^2 - \kappa)}.
\end{aligned} \quad (78)$$

**Lemma 23.** *Let $\overline{B}_1(\sigma, \kappa)$ be given by (76). Then, for any $\kappa \geq 2$ and all $\sigma \geq 0$, $\overline{B}_1(\sigma, \kappa) \geq 0$.*

*Proof.* Let $s(\sigma) = 1 + \tau_0$. By the definition (76), we can write

$$\overline{B}_1(\sigma, \kappa) = -\frac{\tau_0 N_{\overline{B}_1}(s(\sigma), \kappa)}{\kappa s(\sigma)^3}, \qquad N_{\overline{B}_1}(s, \kappa) := s^2(3 - s) - \kappa(s + 1).$$

From the construction we have $s(\sigma) \geq \kappa$ and here we assume $\kappa \geq 2$, so in particular $s(\sigma) \geq 2$. Since $\tau_0 > 0$, $\kappa > 0$ and $s(\sigma) > 0$, the prefactor $-\frac{\tau_0}{\kappa s(\sigma)^3} < 0$. Therefore, to prove $\overline{B}_1(\sigma, \kappa) > 0$ it suffices to show

$$N_{\overline{B}_1}(s, \kappa) < 0 \qquad \text{for all } s \geq \kappa \geq 2.$$

We analyse $N_{\overline{B}_1}$ as a function of $s$ (with $\kappa$ fixed). Its derivatives are

$$N'_{\overline{B}_1}(s, \kappa) = 6s - 3s^2 - \kappa, \qquad N''_{\overline{B}_1}(s, \kappa) = 6 - 6s.$$

For $s \geq 2$ we have $N''_{\overline{B}_1}(s, \kappa) < 0$, so $N_{\overline{B}_1}$ is concave on $[2, \infty)$, and hence on $[\kappa, \infty)$ since $\kappa \geq 2$. We first evaluate $N_{\overline{B}_1}$ and its derivative at the boundary point $s = \kappa$:

$$N_{\overline{B}_1}(\kappa, \kappa) = \kappa^2(3 - \kappa) - \kappa(\kappa + 1) = -\kappa(\kappa - 1)^2 < 0,$$

and

$$N'_{\overline{B}_1}(\kappa, \kappa) = 6\kappa - 3\kappa^2 - \kappa = \kappa(5 - 3\kappa).$$

For $\kappa \geq 2$ we have $5 - 3\kappa < 0$, so $N'_{\overline{B}_1}(\kappa, \kappa) \leq 0$. Since $N_{\overline{B}_1}$ is concave on $[\kappa, \infty)$, its derivative $N'_{\overline{B}_1}(s, \kappa)$ is non-increasing in $s$ on this interval. Hence, for all $s \geq \kappa$, $N'_{\overline{B}_1}(s, \kappa) \leq N'_{\overline{B}_1}(\kappa, \kappa) \leq 0$, so $N_{\overline{B}_1}(\cdot, \kappa)$ is non-increasing on $[\kappa, \infty)$. Together with $N_{\overline{B}_1}(\kappa, \kappa) < 0$ this implies

$$N_{\overline{B}_1}(s, \kappa) \leq N_{\overline{B}_1}(\kappa, \kappa) \leq 0 \qquad \text{for all } s \geq \kappa \geq 2,$$

which proves the lemma. $\qquad\square$

**Lemma 24.** *Let $\overline{B}_2(\sigma, \kappa)$ be given by (78). Then, for every $\kappa > 1$ and all $\sigma \geq 0$, $\overline{B}_2(\kappa, \sigma) \geq 0$.*

*Proof.* Recall the definition

$$\overline{B}_2(\sigma, \kappa) = \frac{2\tau_0^2}{(1 + \tau_0)((1 + \tau_0)^2 - \kappa)}.$$

Both the numerator and the denominator are positive for $\kappa > 1$ and $\sigma > 0$ (as shown in the previous lemmas), therefore the claim holds. $\qquad\square$

**Lemma 25.** *Let $\overline{C}_2(\sigma, \kappa)$ be given by (78). Then, for every $\kappa \geq 2$ and all $\sigma \geq 0$, $\overline{C}_2(\kappa, \sigma) \geq 0$.*

*Proof.* Recall the definition

$$\overline{C}_2(\sigma, \kappa) = \frac{2\tau_0^3(\tau_0^2 - 1)}{(1 + \tau_0)^4((1 + \tau_0)^2 - \kappa)}.$$

Let $s(\sigma) = 1 + \tau_0$. The denominator is strictly positive for $\kappa > 1$. The term $2\tau_0^3$ is also strictly positive. Thus, the sign of $\overline{C}_2$ is determined by the sign of $(\tau_0^2 - 1)$. We rewrite this term as:

$$\tau_0^2 - 1 = (s(\sigma) - 1)^2 - 1 = s(\sigma)^2 - 2s(\sigma) = s(\sigma)(s(\sigma) - 2).$$

Since $s(\sigma) \geq \kappa > 1$, $s(\sigma)$ is positive. The sign is therefore determined by $(s(\sigma) - 2)$. We are given $\kappa \geq 2$, which gets $s(\sigma) \geq \kappa \geq 2$. Therefore, $s(\sigma) - 2 \geq 0$ for all $\sigma \geq 0$ and the claim holds. □

## E  DETAILS FOR THE EXPERIMENTAL SETUP

For both datasets, the curves are obtained by running 100 equally spaced values of $\lambda$ with the same splits, so that the observations focus on the performative effect. Data is split uniformly at random across the different steps.

**Housing.**  We keep all features of the dataset and normalize them. We center the target feature since we use a linear regression without intercept. Following Cyffers et al. (2024), we fix the features affected by the performative effect to be `MedInc`, `AveBedrms`, and `AveOccup`, with all values of $b$ set equal.

**LSAC.**  We keep only one feature in cases of redundant encoding, drop features that are too strongly correlated with the target `GPA` ($\rho > 0.6$), and randomly select roughly half of the features to be affected by the performative effect. The names of these features are reported in Table 1. We normalize all features and center the target. All coefficients of $b$ are equal. In Figure 5, we observe that the features do not follow the assumptions done on $X$ in the theoretical part, despite exhibiting similar behavior in the experiments.

Table 1: Features of the LSAC dataset

| Category | Feature name |
|----------|-------------|
| Redundant | `male` (same as `sex`), `parttime` (same as `fulltime`), `decile1` (same as `decile1b`) |
| With $\rho > 0.6$ | `ugpa`, `index6040`, `dnn bar pass prediction` |
| With $b_{\text{feat}} = \bar{b}$ | `Unnamed0`, `decile1b`, `decile3`, `other`, `asian`, `black`, `hisp`, `pass bar`, `tier` |

**Reproducibility and LLM Usage**  We plan to publicly release the code upon acceptance. All experiments were run on a single laptop. Large Language Models were used only for rewording, formatting, basic scripting and documentation.

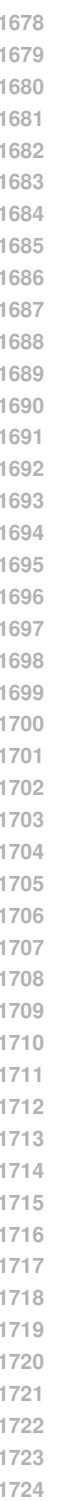

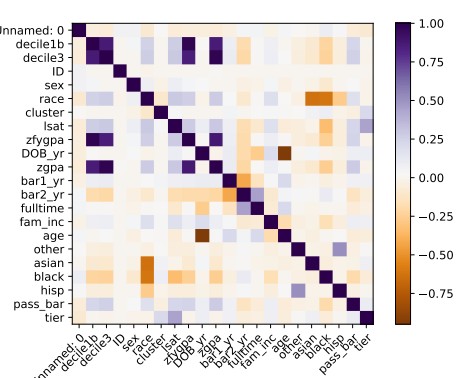

Figure 5: Empirical covariance of LSAC dataset.

