# OpenReview forum: "Optimal Regularization for Performative Learning"
_ICLR.cc/2026/Conference — Submitted to ICLR 2026_

### Official Review · Reviewer_foLY · 2025-10-26

**Soundness:** 2
**Presentation:** 2
**Contribution:** 1
**Rating:** 2
**Confidence:** 4

**Summary:**

This paper investigates the interaction between regularization and performative effects in performative learning, a framework where the data distribution adapts to the deployed model (e.g., through user behavior or feedback loops). The authors focus on ridge regression and study how the optimal regularization parameter should be tuned to mitigate performative shifts. The analysis is conducted under strong modeling assumptions: the data-generating process is fully Gaussian, the conditional model for $y \mid x$ is linear with both predictive and spurious features, and the performative shift acts linearly on the covariates. Within this stylized setting, the paper characterizes how the optimal regularization depends on the strength and direction of the performative effect.

**Strengths:**

The paper provides an analysis of how regularization interacts with performative effects in ridge regression. It offers closed-form expressions for the optimal regularization in population regimes (small number of covariates and over-parameterized), supported by empirical validation.

**Weaknesses:**

1) **Modeling assumptions are overly restrictive** and cast doubt on the generality of its conclusions. The entire analysis is built on a fully Gaussian linear framework, where:
- the performative effect acts only on the covariates and the dependence is linear;
- the conditional model for $y \mid x$ is linear with additive Gaussian noise; and
- The covariates are mean-zero Gaussian.

This highly stylized formulation makes the mathematics tractable but reduces conceptual novelty. More importantly, my main concern about this specific model's assumptions is that the proportionality between the optimal ridge parameter and the performative strength follows directly from the linear–Gaussian algebra rather than from a general property of performative learning. Consequently, it is unclear whether the claimed insights extend to nonlinear predictors, heteroskedastic noise, or performative effects that act on the label rather than on the covariates, or even to more realistic yet still structured settings such as those involving strategic agents.

2) **The paper’s relationship to prior work (Perdomo et al., 2020)** is somewhat misrepresented.

The authors claim their model generalizes the performative regression model of Perdomo et al.; however, the opposite is true.
The setup in this paper is a special case of the general framework in Perdomo et al.:
- their model corresponds closely to *Example 2.2* in Perdomo et al., where the performative shift is encoded through a linear transformation of the covariates. The only substantive difference from *Example 2.2* in Perdomo is the inclusion of spurious covariates and that in Perdomo, $y$ is binary, and in this work, it is continuous.
- in contrast, Perdomo et al. allow a general distributional shift in the joint $(X,Y)$, while this paper restricts the shift to $X$ only;
- the outcome $Y$ in Perdomo et al. is not required to be linear or Gaussian, whereas here it is; and
- Perdomo et al. consider general loss functions, while this paper is limited to the squared loss.

Hence, the contribution should be regarded as an analytical refinement of a specific subcase within Perdomo’s framework, rather than as a generalization.

3. **Recent work in high-dimensional setting**. The paper also overlooks recent works that already study high-dimensional performative settings without relying on such restrictive Gaussian-linear models and squared loss assumptions. For instance, [1] analyzes regret minimization under general performative feedback, and their convergence only depends on the zooming dimension (which can be much smaller than the parameter dimension); and [2] establishes dimension-independent convergence results in the strategic setting with general loss. These works demonstrate that meaningful high-dimensional analysis is possible under more general conditions.

## References
[1] Jagadeesan, Meena, Tijana Zrnic, and Celestine Mendler-Dünner. "Regret minimization with performative feedback." International Conference on Machine Learning. PMLR, 2022.

[2] Bracale, Daniele, et al. "Learning the distribution map in reverse causal performative prediction." arXiv preprint arXiv:2405.15172 (2024).

**Questions:**

1. Do you also assume $\mathbb{E}( \theta^* )=0$ for Equation (7) to hold? The derivation seems to require the population parameter to be centered; otherwise, the cross-term $\mathbb{E}( (\theta^* )^\top A^T \Sigma A \theta^* )$ would not simplify as stated.

2. Why do you introduce the empirical version of $\theta_k$ in Eq. (3), (13) and (14) when the rest of the analysis is conducted entirely at the population level? This switch in notation could confuse readers, as it is unclear whether finite-sample randomness ever plays a role in the theoretical results.

3. In the abstract, you state:
   > “We show that, while performative effects worsen the test risk in the population setting, they can be beneficial in the over-parameterized regime.”

However, it seems that it is not the *performative effect itself* that is beneficial, but rather that in the presence of performativity, optimal regularization helps reduce the variance or uncertainty of the estimated parameters, thereby improving performance. Could you clarify or restate this claim to better reflect the mechanism driving the improvement?

4. After Equation (17), you refer to $\widehat{ R  } _ {eq}$, but I don't find it defined in the text.  Could you indicate where this expression is formally introduced?

5. The performative effect is modeled as a diagonal linear transformation. Would the analysis still hold if $D$ had off-diagonal entries, i.e., if the performative shift mixed predictive and spurious features?

---

> ### Author Response · Authors · 2025-11-20
> **Answers to the weaknesses**
>
> We thank the reviewer for the comments. We address all concerns and questions below, one by one. We have edited our paper accordingly and uploaded a revised version, where the changes are marked in red color to make them more visible.
>
> **Restrictive assumptions** Our assumptions, albeit restrictive, are common in the theoretical literature on high-dimensional regression, see the paragraph **High-dimensional regression and role of ridge regularization** and references therein.
>
> We note that the Gaussian assumption on the covariates could be relaxed. In fact, the analysis in the population setting of Section 4 does not require this assumption. The analysis in the over-parameterized setting of Section 5 builds on results from (Han & Xu, 2023) and more precisely on their Theorem 2.3. Such result is already extended in Theorem 2.4 of (Han & Xu, 2023) to the case in which the whitened covariate $\Sigma^{1/2}x$ has independent, zero mean, unit variance, and uniformly subgaussian entries. Hence, a similar adaptation should carry over to our setup at the cost of a more involved analysis. We now mention this in l. 346-350 of the revision.
>
> Let us conclude by mentioning that the numerical results of Section 5 hold for real-world datasets which are neither Gaussian nor linear, confirming the validity of such assumptions. We added in Appendix E the covariance of the real data used, so it is easy to check that it is far from the theoretical setting. The fact that high-dimensional results are often established under a rather restrictive setting but still give correct predictions when the hypotheses are not satisfied is one of the explanations of the success of this field and is not specific to our work.
>
> **Prior work discussion**
> We have added in the revision that we were comparing ourselves to Example 2.2 of (Perdomo et al., 2020). We agree that (Perdomo et al., 2020) define more general quantities as they define performative learning in general. However, Example 2.2 is the only one that is studied through the whole paper, and the only one for which $\theta_{PS}$ or $\theta_{PO}$ is computed. We do not claim to cover all performative learning scenarios, we made the connection to this prior work to show that this model was already seen as an interesting scenario, even in a less general version, in highly-cited previous work.
>
> **Related work** We thank the reviewer for pointing out these two relevant works, we added them to our related work section. We would like to stress that these two works mainly show the growing interest for connecting high-dimensional settings and performative learning, but address different topics. In [1], the goal is to minimize the regret which means the whole trajectory, rather than the final stable points; the developed algorithm requires to be able to estimate confidence bounds performance on models that have not been deployed with respect to a distribution which was deployed; none of the tools of high-dimensional regression is used, and what appears is, as you said, the $\alpha$-zooming dimension, which depends on the Lipschitzness of the data distribution and the loss; and finally, no setting close to the proportional one we are interested in is tackled. In [2], the authors take a causal perspective mainly for finite action states, which is not exactly our focus here, but indeed the paper is another proof that anticipating distributions shift is a complex issue. Having a simple way to agnostically regularize in order to mitigate performative effects is thus an interesting complementary approach.

---

> ### Author Response · Authors · 2025-11-20
> **Answers to the questions**
>
> *Question 1. Do you also assume $\mathbb{E}(\theta^{\star})=0$ for Equation (7) to hold?*
>
> **Response.** The reviewer is correct. By assumption 1 $(\theta_{pop}^{*})^{\top}$ is exactly $(a^{\top}, 0)^{\top}$,and $a$ has zero mean, which implies that $\mathbb{E}(\theta^{\star}_{pop})=0$. We have clarified this in l. 129 of the revision.
>
> *Question 2. Why do you introduce the empirical version of $\theta_k$ in Eq. (3), (13) and (14) when the rest of the analysis is conducted entirely at the population level? This switch in notation could confuse readers, as it is unclear whether finite-sample randomness ever plays a role in the theoretical results.*
>
> **Response.** We believe there is a misunderstanding here. **The analysis in Section 5 is not conducted at the population level**, and finite-sample randomness does play a role in the theoretical results, which otherwise would be the same as those in Section 4 (which indeed deals with the population case). In Section 5, both the sequence $(\theta_k)_k$ and the corresponding risk are random, which makes the analysis challenging. To address this challenge, Theorem 3 provides a deterministic equivalent to the risk (holding with high probability over the data).
>
> This misunderstanding is also echoed in the description of the strengths of our paper: “*It offers closed-form expressions for the optimal regularization in population regimes (small number of covariates and over-parameterized)*”. We believe this is a critical point: **the core technical contribution of our work consists precisely in going beyond the population regime and analyzing the challenging over-parameterized regime where the number of samples scales linearly with the dimension of the features**. We hope that this clarification leads the reviewer to reconsider the assessment of our work.
>
> *Question 3. In the abstract, you state: “We show that, while performative effects worsen the test risk in the population setting, they can be beneficial in the over-parameterized regime.” However, it seems that it is not the performative effect itself that is beneficial, but rather that in the presence of performativity, optimal regularization helps reduce the variance or uncertainty of the estimated parameters, thereby improving performance. Could you clarify or restate this claim to better reflect the mechanism driving the improvement?*
>
> **Response.** We agree and have restated the claim according to the suggestion of the reviewer: “We show that, while performative effects worsen the test risk in the population setting, when moving to the over-parameterized regime where the number of features exceeds the number of samples, the optimal regularization in the presence of performativity helps reduce the variance in the estimated parameters, thereby improving performance.”
>
> *Question 4. After Equation (17), you refer to $\hat{R}_{eq}$, but I don't find it defined in the text. Could you indicate where this expression is formally introduced?*
>
> **Response.** Thank you for spotting this typo. We meant $R_{eq}$ and have corrected the revision accordingly (see l. 337).
>
> *Question 5. The performative effect is modeled as a diagonal linear transformation. Would the analysis still hold if $D$ had off-diagonal entries, i.e., if the performative shift mixed predictive and spurious features?*
>
> **Response.** The general result of Theorem 3 holds for a general matrix $D$ (which could have non-zero off-diagonal entries). This is now clarified in l. 346-350 of the revision. To obtain the more precise analysis in Theorem 4, we resort to the assumption that $D$ is diagonal. We are not aware of previous work interpreting non-diagonal coefficients; in particular (Cyffers and al. 2024, Izzo et al., 2022; Hardt & Mendler-Dünner, 2023) also only rely on diagonal coefficients. We specify in the following theorems when the fact that $D$ is diagonal is needed. Intuitively, diagonal coefficients can be interpreted directly as the modifications made by a strategic agent, depending on how the feature is used and the cost of modifying it. We added this explanation to the paper and clarified when we use the diagonal hypothesis.
>
> Let us finally point out that, even if $D$ is diagonal, predictive and spurious features are still mixed via the covariance $\Sigma$ which is not diagonal. We have edited the revision in l. 376-377 to clarify these points.
>
> We thank Reviewer foLY for this detailed review. We believe that the modifications addressed all the raised concerns and hope that the revised version will be evaluated positively. We will be happy to continue the discussion if some points remain unclear.

---

> > ### Comment · Reviewer_foLY · 2025-11-26
> > **Sub-Gaussian Covariates**
> >
> > I thank the author for the response and for the careful and thoughtful explanation. I appreciate the effort to clarify all the points I raised. In what follows, I will discuss the assumption of Gaussianity of the covariates further. In the updated version, you write:
> >
> > *The assumption on the features $x$ being Gaussian could be also relaxed at the cost of a more involved analysis. In fact, the results of Han \& Xu (2023) (see Theorem 2.4 therein) hold for $\Sigma^{-1 / 2} x$ having independent, zero mean, unit variance and uniformly subgaussian entries.*
> >
> > However, this remark may leave the reader uncertain about how Theorem 2.4 from Han \& Xu (2023) actually connects to the arguments developed in the paper. The theorem is certainly relevant, but without further explanation, it is not immediately clear how it would interact with the key steps in the proofs-particularly those in Sections 4 and 5 where the Gaussianity of the covariates is explicitly used. I understand that adding a full theoretical supplementary section explaining extension to non-Gaussian covariates may not be feasible at this stage. However, this might be actually necessary because, to the best of my limited knowledge in this topic, relaxing Gaussianity of the covariates typically requires additional technical assumptions beyond mere sub-Gaussianity. For example, Condition B. 54 in [1] illustrates how more delicate structural assumptions on the design distribution can become necessary when working outside the Gaussian setting. Without clarifying whether similar conditions would be needed here-and if so, how they would propagate through the proofs-the reader may find it difficult to assess whether Theorem 2.4 from Han \& Xu (2023) alone is sufficient to support the generalization.
> >
> > This would strengthen the paper, particularly because the experiments already demonstrate that non-Gaussian covariates still yield behavior consistent with the theory. I am raising my rating a bit for now.
> >
> >
> > ## References
> >
> > [1] Bombari, S. and Mondelli, M., 2025. Spurious Correlations in High Dimensional Regression: The Roles of Regularization, Simplicity Bias and Over-Parameterization. arXiv preprint arXiv:2502.01347.

---

> > > ### Author Response · Authors · 2025-11-26
> > > **We added Sub Gaussian Covariates version**
> > >
> > > We thank the reviewer for engaging in a productive discussion and for the helpful suggestion.
> > >
> > > In short, we agree: having a complete proof of the extension to sub-Gaussian covariates would strengthen the paper. So, we have added this extension in Appendix B.1 of the new revision that we have just uploaded.
> > >
> > > As correctly mentioned by the reviewer, the key step consists in showing that Condition B.54 in [1] holds for $\theta_0, \theta_{\rm pop}^{\star}, \theta_1$.
> > > In addition to that, the new argument also needs a bound on $\lVert \theta_1\rVert_2, \lVert \theta_2\rVert_2$. We prove such bounds in Lemma 10 of the revision. We note that we don’t need to make any additional delicate structural assumption on the setup: we just assume that $\theta_0$ is sampled uniformly on the unit sphere (which is mild, since we can choose the initialization of the algorithm) and that $a$ (and, therefore, $\theta_{\rm pop}^*$) is sub-Gaussian (which is also mild).
> > >
> > > At this point, the proof of the extension of Theorem 3 to sub-Gaussian features (now Theorem 9 in the revision) follows using basically the same argument of Theorem 3. This is detailed in Appendix B.1.
> > >
> > > We hope that our new revision clarifies completely the issue raised by the reviewer and, as a consequence, that the reviewer considers raising further the score.

---

> > > > ### Author Response · Authors · 2025-11-28
> > > >
> > > > Dear Reviewer,
> > > >
> > > > Thank you again for suggesting relaxing the Gaussian data hypothesis. As we have updated the paper accordingly, and as we understand this was your only remaining concern, could you update your position regarding our paper?
> > > >
> > > > Best,
> > > > The Authors

---

### Official Review · Reviewer_XghE · 2025-10-27

**Soundness:** 3
**Presentation:** 1
**Contribution:** 3
**Rating:** 4
**Confidence:** 2

**Summary:**

This paper investigates what happens for linear regression applied in performative settings (where data reacts to the deployed model) when the model is optimally regularized via L2 regularization. Both the under parametrized and overparametrized are studied theoretically.

**Strengths:**

The theory agrees with the numerical experiments

It makes sense to study a technique like regularization for this setting

**Weaknesses:**

The paper has very low self-containment and lacks intepretation. As someone who is vaguely familiar with performative prediction, I could not understand the setting from reading this paper. For example; Assumption 1 introduces various variables, such as a, b, c, d. Yet their interpretation is not mentioned. Also $a$ seems to have a covariance; the interpretation for me is not clear. Are we then in a Bayesian setting and is there a prior on $a$? Or are we in a frequentist setting and we do typically consider worst-cases with respect to $a$? Because the setting is not clearly introduced, this really hampered my reading of the whole paper.

The writing is very technical; I cannot get the main points easily. Even the Figure captions are so technical, with very little interpretation, I cannot understand their points.

**Questions:**

"In this section, we tackle the population regime where there are enough samples from D(θk) at each deployment to compute exactly the next regressor, as would typically happen in a low-dimensional setting."

How is this possible in the presence of noise?

---

> ### Author Response · Authors · 2025-11-20
>
> We thank the reviewer for the comments and answer below. We have edited our paper and uploaded a revised version, where the changes are marked in red color to make them more visible.
>
> **Explaining $a$** In this paper, we stand at the intersection of performative learning and high-dimensional regression, and we agree that it might be difficult to follow the technical details when unfamiliar with high-dimensional statistics. In the specific case of $a$, we assume that $a$ is drawn from a distribution with zero mean and covariance $I_d/d$ (you may think of a Gaussian distribution, for instance, if you prefer to fix a concrete distribution). This is not a limitation, as all directions in space are covered and any scaling can be handled through pre-processing of the data. Assuming that the vector is drawn from a distribution is then used to interpret the risk as a trace, which is a standard technique in high-dimensional statistics.
>
>
> **Population setting explanation** The population regime corresponds to the setting where one has many samples $n$ relative to the dimension $d$. In this scenario, the individual noises cancel each other while the signal adds up, which allows concentration results to hold. In particular, for this specific case, the empirical covariance matrix $\frac{1}{n} X^\top X$ concentrates to the covariance matrix $\Sigma$, and this is why we can make the substitution.

---

### Official Review · Reviewer_QCgE · 2025-10-31

**Soundness:** 3
**Presentation:** 3
**Contribution:** 3
**Rating:** 6
**Confidence:** 3

**Summary:**

The paper investigates how regularization can mitigate performative effects, e.g., situations where the data distribution depends on the deployed model. The authors focus on high-dimensional ridge regression and study both the population regime and the over-parameterized regime. In particular, it shows that 1) in the population regime, the optimal regularization is proportional to the magnitude of the performative effect and can mitigate performance degradation; 2) in the over-parameterized regime, performative effects can actually improve test risk when properly regularized. The authors derive closed-form characterizations of the optimal ridge coefficient and deterministic equivalents for performative risk, supported by both synthetic and real-world experiments (Housing, LSAC datasets).

**Strengths:**

Overall, the paper is well-written and well-structured. It provides one of the first analytical treatments of regularization under performativity, linking performative dynamics to the scaling of optimal regularization. The mathematical contributions are rigorous, and the main theorems (Theorem 1, 3, 4) are clearly stated. The finding that performative effects can improve performance in the over-parameterized regime (contrary to intuition) is conceptually interesting and could have a broader impact on regularization with the presence of performativity.

**Weaknesses:**

My biggest concern with the paper is that it relies on restrictive modeling assumptions: in particular, the analysis assumes Gaussian features and linear label shifts (Assumption 1). This limits generalizability to nonlinear or heavy-tailed data distributions, which are common in real-world performative settings. In addition, the paper only focuses exclusively on $\ell_2$ regularization; other forms (ℓ₁, dropout, early stopping) are only mentioned in future work but not studied.

While the theoretical results are rigorous and technically sound, many expressions (e.g., Theorem 3 and its expansions) are algebraically heavy and include higher-order terms that obscure intuition. Even though the authors provide closed-form expressions, the results are not immediately interpretable without significant algebraic unpacking. A more intuitive discussion or simplified special cases (e.g., isotropic $\Sigma$ etc) would help readers understand the qualitative behavior of the optimal regularization.

**Questions:**

In Eq. (4), the excess risk is defined as the test risk under the unshifted distribution $D(\theta = 0)$. While this isolates the performative effect and prevents evaluation bias, it seems somewhat counterintuitive to me; in reality, a deployed model is evaluated on the induced distribution $D(\theta^∗)$. Could the authors clarify the motivation for this evaluation choice and discuss how the conclusions might differ if the test risk were computed on $D(\theta^∗)$ instead?

---

> ### Author Response · Authors · 2025-11-20
>
> We thank the reviewer for the positive evaluation of our work and for the comments. We address the questions below. We have edited our paper accordingly and uploaded a revised version, where the changes are marked in red color to make them more visible.
>
> **We added discussion on why we evaluate on $D(\theta=0)$** First, let us point out that the choice of testing under the unshifted distribution $\mathcal D(\theta=0)$ starts to be popular as well. In fact, it can be interpreted as evaluating the model on the untouched distribution, and the shift due to performativity can be interpreted as a bias reinforcing itself in real-world scenarios. Evaluating the performance on the unshifted distribution is thus a way to know how prone the model is to produce a more unfair world. A simple example was already studied in [1,2]. For a classification task, if the performative effect reinforces the probability of collecting data points from the easy-to-learn classes, when evaluating models on the induced distribution, it means aiming for the scenario where only one class remains, as the constant predictor is then optimal and archives perfect accuracy, which is not an interesting goal. Similarly, in [2], the fact that the placebo effect increases with drug effectiveness is still seen as an undesirable effect, and the goal is to predict the “true” effect. In [1,2], the authors thus also choose to test on the untouched distribution (uniform across labels).
>
> [1] Adjusting Pretrained Backbones for Performativity, Berker Demirel, Lingjing Kong, Kun Zhang ,Theofanis Karaletsos, Celestine Mendler-Dünner and Francesco Locatello
>
> [2] On the Impact of Performative Risk Minimization for Binary Random Variables, Nikita Tsoy, Ivan Kirev and Nikola Konstantinov
> Second, as we test how well regularization works for taming performative learning when there are also spurious features, a starting point of this study was to test whether or not we would lose the benefit of regularization because of amplified spurious correlations. Thus, it was important to test on the initial distribution, where the spurious features are indeed spurious. The main take-away of our work is that the drawback of reinforcing spurious correlations is very limited (as it appears only at the third order) in comparison to the benefit we have for mitigating performativity. This could not have been measured by testing in-distribution.
>
> Third, anticipating the next question, we started this study with the population regime, where performative effects do not modify the behavior of the distribution. In fact, for all deployed $\theta$, it is possible to learn exactly the current model, achieving an excess risk of zero. What is more interesting to study is how the excess risk evolves across time, and thus how we perform with respect to the first distribution.
>
> We added a condensed version of this answer in Section 3.
>
> **We added to the paper the results on $\mathcal D(\theta^*)$** In the population regime, if we evaluate with respect to the distribution $\mathcal D(\theta^{\star})$ (i.e., the same distribution according to which the samples are drawn), the excess risk is equal to 0 at every iteration, see l. 185-186 of the paper. This means that the problem effectively trivializes.​​
> The problem is however non-trivial in the over-parameterized setting considered in Section 5. To address the question posed by the reviewer, using the same approach developed to prove Theorems 3-4, we have obtained results for testing with respect to $\mathcal D(\theta^{\star})$. More precisely, Theorem 16 is the equivalent of Theorem 3 and it provides a deterministic equivalent of the performative fixed point. Lemma 17 specializes the risk expression for testing with respect to $\mathcal D(\theta^*)$ to the case in which the covariance has the form $\Sigma=\begin{bmatrix}I_d&\rho I_d\\ \rho I_d&I_d\end{bmatrix}$, and it is the equivalent of Lemma 9. Next, we provide expressions for the optimal $\tau$ in Lemma 18, for the optimal regularization parameter in Equations (65)-(66), and for the optimal risk in Corollary 19. Lemma 20 shows that the optimal regularization moves in the same direction as the performative effect on the predictive features, as it is the case for testing over $\mathcal D(\theta=0)$ provided that the noise variance is small. Finally, Lemma 21 and 22 show that the optimally regularized risk worsens in the presence of a performative effect (either on the predictive or on the spurious features) that reinforces existing trends ($\bar b, \bar c>0$). This is in contrast with the behavior of the optimally regularized risk tested over $\mathcal D(\theta=0)$ which instead decreases when either $\bar b>0$ or $\bar c>0$. All these new results are stated and proved in Appendix D of the revision, and they are discussed in l. 447-461 of the body of the revision.

---

> > ### Author Response · Authors · 2025-11-28
> >
> > Dear Reviewer,
> >
> > Your initial review is positive and listed one main concern, the evaluation on $\mathcal{D}(\theta = 0)$ rather than on $D(\theta = \theta^{\star})$. As our revision added the results for the evaluation on $\mathcal{D}(\theta = \theta^{\star})$, can we hope that you will raise your score further and champion our paper toward acceptance?
> >
> > Best,
> >
> > The Authors

---

### Official Review · Reviewer_gNfJ · 2025-11-01

**Soundness:** 4
**Presentation:** 3
**Contribution:** 1
**Rating:** 2
**Confidence:** 2

**Summary:**

This submission considers "performative prediction" setting introduced by Perdomo et al. (2020). Performative prediction is a learning setting where a decision-maker repeatedly deploys a model and nature responds with a slightly altered distribution. A key example is when individuals may strategically alter their features in order to receive better classification under the current predictive model.

This submission works in the setting where the decision-maker uses ridge regression; it focuses on exploring the interaction of regularization and performativity. The authors prove several theorems about the convergence and excess risk of this process. They also run some experiments.

Mathematically, after the decision-maker deploys a model $\theta$, the data arise from distribution $D(\theta)$ as follow: $x_i \sim \mathcal{N}(0,\Sigma)$ and $y_i \sim x^T \theta^* + x^T D \theta + \mathcal{N}(0,\sigma^2)$. Here $D$ is a matrix that controls the performative effects. The sequential retraining process may converge to some $\theta^{\infty}$. The theorems evaluate performative risk with respect to $D(\theta=0)$. In contrast, the prior work I am familiar with evaluates excess risk on $D(\theta^{\infty})$, i.e., at the equilibrium we actually reach.

**Strengths:**

I think this is an interesting area in which to work. I found the paper polished and easy to read.

The issue I address below may not be fundamental, in which case I would view the paper favorably.

**Weaknesses:**

As mentioned above, this paper evaluates excess risk on the $D(\theta=0)$ distribution, which in general will not be the equilibrium distribution. In contrast, the work on performative prediction I am familiar with evaluates risk on the equilibrium distribution (see [1]). It is not clear to me why the results here are of interest.

The authors provide some reasoning: "This ensures that the final model is not evaluated on shifted distributions, and it is particularly relevant for long-term fairness, as it prevents bias amplification over time... and discourages reliance on spurious feature." I do not understand how these address the concern above.

However, I am not an expert in the area so I look forward to discussion with the authors and other reviewers.

[1] Hardt, Moritz, and Celestine Mendler-Dünner. "Performative prediction: Past and future." Statistical Science 40.3 (2025): 417-436. [https://arxiv.org/abs/2310.16608](https://arxiv.org/abs/2310.16608)

**Questions:**

Why did you choose to evaluate excess risk with respect to $D(\theta=0)$?

How do the results change if you evaluate with respect to $D(\theta^*)$?

---

> ### Author Response · Authors · 2025-11-20
>
> We thank the reviewer for the comments and notably for noting that the paper was easy to read. We address the two questions below. We have edited our paper following your remarks and uploaded a revised version, where the changes are marked in red color to make them more visible.
>
> **Choice to evaluate on $\mathcal D(\theta=0)$**
> Our choice was first motivated by three reasons, that we detail in the answer to Reviewer QCgE
>  who asked the same question. In summary, this choice has already been used in previous works, is necessary to evaluate the impact on spurious correlation, and evaluation in-training is less interesting for the population setting. We added this discussion to the paper.
>
> **Results on $\mathcal D(\theta^*)$** We also answer in detail in our response to Reviewer QCgE. We have added the results to the paper for completeness. The excess risk always remains null for the population setting, but exhibits more complex behavior in the proportional setting.
>
> We thank the Reviewers gNfJ and QCgE for pointing out these limitations on the previous version of our work. We hope that the revised version, which fixes this weakness, will modify your point of view on our paper, so that you can support acceptance.

---

> > ### Comment · Reviewer_gNfJ · 2025-11-21
> >
> > Thank you for the response. I've looked at the updated submission, the other reviews, and your responses, and I did not really update my opinion of the paper. I still feel like I'm lacking a convincing picture of why these results are interesting, to put it bluntly. I have no issues with the technical content, I just feel like it's lacking a motivating story.
> >
> > I'm also a little concerned by Reviewer's foLY's notes about related work. I'm not fully convinced by your response to them. For example, you say that regret is different as it considers all time steps instead of the equilibrium point, but in the overparameterized setting your analysis seems to rely on the fact that the process has essentially converged after 2 time steps. So it seems to me that the two results might be comparable.

---

> > > ### Author Response · Authors · 2025-11-22
> > >
> > > We thank you for engaging in the discussion with us. We are aware that it is uncommon to change initial reviewer opinion, but we hope this can be one of the exceptions to the rule, as you already agree that we fixed all your technical concerns.
> > >
> > > You cite now only two concerns to recommend reject:
> > > - **the relation to the two papers mentioned by Reviewer foLY**. Let us clarify further that while we agree that these papers hint towards the direction of studying the impact of the dimension in performative learning, they are very distant from our work: they do not study regularization, and they do not study high-dimension settings either.
> > > Their goal is also to study the dependence of the convergence in $T$ when looking at the regret. As you pointed out, this is not something that we could study in the high dimension setting, where only the two steps are important. Let's stress that the algorithms not only do not at all answer the question of regularization, (the word does not even appear in those two papers), and they cannot be applied to our setting. In [1], the main theorem is:
> > >
> > > > Theorem 1.1 (Informal). Suppose that the distribution map $\mathcal{D}(\theta)$ is $\epsilon$-Lipschitz and that the loss $\ell(z ; \theta)$ is $L_z$-Lipschitz in $z$. Then, after $T$ deployments, the performative confidence bounds algorithm achieves a regret bound of
> > > $$
> > > \operatorname{Reg}(T)=\tilde{\mathcal{O}}\left(\sqrt{T}+T^{\frac{d+1}{d+2}}\left(L_z \epsilon\right)^{\frac{d}{d+2}}\right)
> > > $$
> > > where $d$ denotes the zooming dimension of the problem.
> > >
> > > Here, $L_z$ is the quadratic loss, which is not Lipschitz. You can see that as the dimension $d$ grows, the bound becomes meaningless. There is no obvious way to bound the zooming dimension in our setting, so the result (which would not give you any information on the optimal regularization) does not give information when $d$ is big.
> > > In [2], one of the assumptions *(Assumption 2.1 (Reverse causal model)) is  "The $\mathcal{D}(Z \mid A ; \theta)$ are not affected by the prediction model $\theta$. "* where $Z = (X, Y)$ and *$A$ is assumed to be finite*. While we could take $A = Y$, we do not satisfy that $Y$ is finite, so this paper's setting does not cover our setting as well, and does not address the same problem.
> > >
> > > - **the overall motivation of the paper.** As pointed out by the reviewer, performative learning is a relevant topic for the community. Existing results do not discuss the impact of regularization while this method is easy to implement, requires no knowledge of the performative structure contrary to existing methods (such as estimating the performative gradients) and works well in practice, both in small and high dimension. We show that it can counter effect performative shift, and characterize precisely its interaction with it. We believe that our paper has a clear take-home message: add regularization and scale it proportionally to the magnitude of the performative effect. We give precise guarantees under classical high-dimension settings, and show that the prescription remains valid experimentally on real-world data. With the revised version, the take-home message is valid whether or not you want to optimize the risk on the initial distribution (which we believe to be the most interesting setting) or the final distribution.
> > >
> > > We thank you again for taking the time to reconsider your evaluation. We will be happy to answer any other question you may have.

---

> > > > ### Author Response · Authors · 2025-11-28
> > > >
> > > > Dear Reviewer,
> > > >
> > > > Thank you for engaging previously in discussion with us. Could you let us know if you had time to read our last answer and, if your concerns have been addressed, raise your score accordingly?
> > > >
> > > > Best,
> > > > The Authors

---

### Meta-Review · Area_Chair_NiFC · 2026-01-14

**Summary:**

The paper proposes a theoretical study of the impact of regularization in performative learning. Some experimental results are presented on toy datasets: gaussian data, and housing data.

The technical contributions of the work are not compelling enough. All of the calculations can be streamlined using standard RMT ideas. The practical impact of the theoretical results (if any!) are not highlighted enough. Thus, i find this work unimpressive both on the theoretical and experimental/empirical front.

Finally, the authors are proposing too many changes. The final manuscript would look almost like a different paper.

**Reviewer Concerns:**

Here are some major recurrent concerns which cannot be resolved without a major rewrite of the manuscript

- Theoretical setting considered is too toy (Gaussian with shifts), and its not clear if it extends to more realistic scenarios
- Very low self-containment and lacks intepretation of technical results.
- Poor coverage of prior work.
- ...

**Reviewer Scores:**

I doubt the reviewers would have changed their minds (most of their reservations run very deep).

---

### Decision · Program_Chairs · 2026-01-26

Reject